# RestoreGrad: Signal Restoration Using Conditional Denoising Diffusion Models with Jointly Learned Prior

**Ching-Hua Lee** [1]  **Chouchang Yang** [1]  **Jaejin Cho** [1]  **Yashas Malur Saidutta** [1]  **Rakshith Sharma Srinivasa** [1]
**Yilin Shen** [1]  **Hongxia Jin** [1]

## Abstract

Denoising diffusion probabilistic models (DDPMs) can be utilized to recover a clean signal from its degraded observation(s) by conditioning the model on the degraded signal. The degraded signals are themselves contaminated versions of the clean signals; due to this correlation, they may encompass certain useful information about the target clean data distribution. However, existing adoption of the standard Gaussian as the prior distribution in turn discards such information when shaping the prior, resulting in sub-optimal performance. In this paper, we propose to improve conditional DDPMs for signal restoration by leveraging a more informative prior that is jointly learned with the diffusion model. The proposed framework, called RestoreGrad, seamlessly integrates DDPMs into the variational autoencoder (VAE) framework, taking advantage of the correlation between the degraded and clean signals to encode a better diffusion prior. On speech and image restoration tasks, we show that RestoreGrad demonstrates faster convergence (5-10 times fewer training steps) to achieve better quality of restored signals over existing DDPM baselines and improved robustness to using fewer sampling steps in inference time (2-2.5 times fewer), advocating the advantages of leveraging jointly learned prior for efficiency improvements in the diffusion process.

## 1. Introduction

Denoising diffusion probabilistic models (DDPMs) (Ho et al., 2020; Sohl-Dickstein et al., 2015) are latent vari-

able generative models consisting of i) the *forward process*, where the original data samples are gradually corrupted by adding Gaussian noise to eventually become a standard normal prior; ii) the *reverse process*, in which a neural network is responsible for recovering the original data from the corrupted samples by learning to sequentially reverse the diffusion process. With their exceptional capabilities of generating high-quality data, DDPMs can be applied to various signal restoration tasks – recovering the missing components in a signal due to contamination (e.g., audio recorded with environmental noises (Lu et al., 2021; 2022; Tai et al., 2023b), images obstructed by various measurement noises (Özdenizci & Legenstein, 2023; Croitoru et al., 2023)), by conditioning the DDPM on the degraded observations.

However, for the diffusion model to adequately learn the reverse process, a large number of training iterations may be required, leading to potentially slow convergence. Such inefficiency was related to the discrepancy between the real data distribution and the accustomed choice of the standard Gaussian prior by Lee et al. (2022). They therefor proposed a simple yet effective approach called *PriorGrad*, aiming to construct a better prior by using rule-based approaches to extract useful information from the conditioner data. However, despite improving performance on some generative speech tasks, handcrafting a "better" prior requires certain knowledge about the data characteristics, and such guidance may not always exist for a given task or application.

In this paper, our main focus is to investigate the question: *Can we systematically obtain a better prior distribution that improves the efficiency of the diffusion generative process?* In other words, we aim to develop a framework of *learning-based* diffusion priors for improved DDPM efficiency. A high-level view of our approach is depicted in Figure 1, where the conditional DDPM (parameterized by $\theta$) samples the latent noise $\epsilon$ from a learned prior distribution estimated by a *prior encoder* $\psi$, which takes the conditioner $\mathbf{y}$ as input. The prior encoder is jointly trained with the DDPM $\theta$ to synthesize the data $\mathbf{x}_0$, and a *posterior encoder* $\phi$ that exploits information from both $\mathbf{x}_0$ and $\mathbf{y}$, to align the prior and posterior distributions. The main idea here is that, if there is a certain correlation between the conditioner $\mathbf{y}$ and

---

[1]Artificial Intelligence Center−Mountain View, Samsung Electronics, Mountain View, CA, USA. Correspondence to: Ching-Hua Lee <chinghua.l@samsung.com>.

*Proceedings of the 42nd International Conference on Machine Learning*, Vancouver, Canada. PMLR 267, 2025. Copyright 2025 by the author(s).

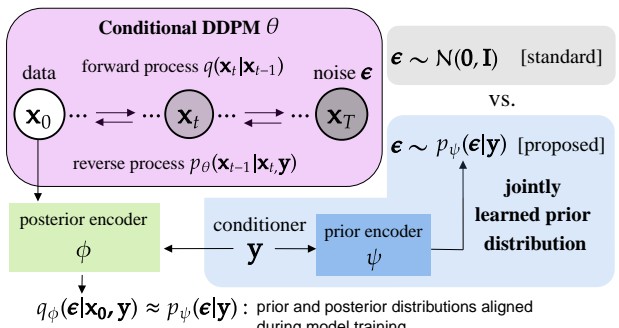

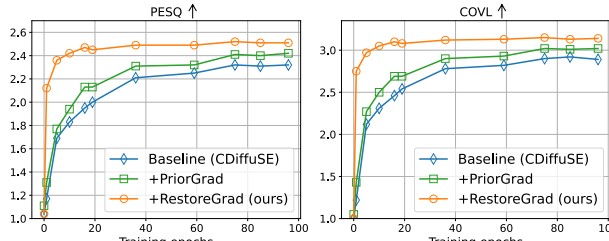

Figure 1. High-level view of the proposed method.

the target data $\mathbf{x}_0$, e.g., in signal restoration problems where $\mathbf{y}$ is typically a degraded version of $\mathbf{x}_0$, our framework can exploit such correlation to construct a more informative prior in an automatic and systematic manner.

To explore the idea, we introduce *RestoreGrad*, a new paradigm for improving conditional DDPM by learning the prior distribution in tandem with the diffusion model, focusing on signal restoration applications. We apply Restore-Grad to speech enhancement (SE) and image restoration (IR) tasks to demonstrate its generality for signals of different nature. For SE, we compare with PriorGrad (Lee et al., 2022) which provides guidance on handcrafting suitable priors in the speech domain. For IR, we show that RestoreGrad serves as a promising solution for improving the baseline DDPM even in a domain that lacks such a recipe for engineering the prior. As shown in Figure 2, models trained using RestoreGrad are more data and compute-efficient than the baseline DDPMs and PriorGrad; they converge faster to achieve higher quality of the restored signal. Further shown in Figure 3, the learned prior is more informative as it better correlates with the desired signal than an isotropic covariance, potentially simplifying the diffusion trajectory for improved model efficiency.

Our main contributions are summarized as follows:

- We study the problem of learning the prior distribution *jointly* with the conditional DDPM for signal restoration applications, aiming at providing a more systematic, *learning-based* treatment to address the inefficiency incurred by existing selections of the prior distribution.

- We propose a new framework called RestoreGrad that learns the prior in conjuncture with the DDPM model through a *prior encoder*, by exploiting the correlation between the targe signal and input degraded signal encoded by an auxiliary *posterior encoder*, for improved model efficiency. Our *two-encoder* learning framework is established based on a novel evidence lower bound (ELBO) that seamlessly integrates the DDPM into the variational autoencoder (VAE) (Kingma & Welling, 2014) to harness the advantages of both methodologies.

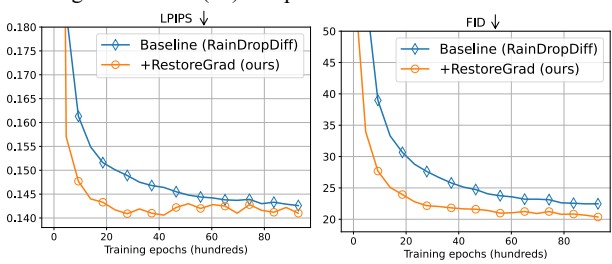

Figure 2. Model learning performance. (Top) In the speech domain, RestoreGrad outperforms PriorGrad (Lee et al., 2022), a recently proposed improvement to the baseline DDPM (CDiffSE (Lu et al., 2022)) which leverages handcrafted priors. (Bottom) In the image domain, RestoreGrad provides a paradigm to improve the DDPM baseline (RainDropDiff (Özdenizci & Legenstein, 2023)) when there is no existing recipe for obtaining better priors.

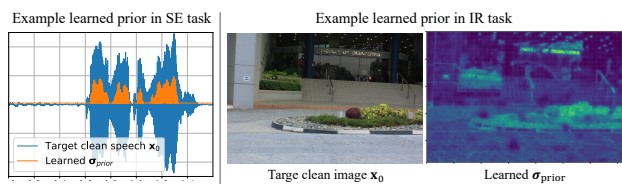

Figure 3. Visualization of the learned prior distribution. Here, the distribution of the prior is modeled as: $p_\psi(\boldsymbol{\epsilon}|\mathbf{y}) := \mathcal{N}(\boldsymbol{\epsilon}; \mathbf{0}, \text{diag}\{\boldsymbol{\sigma}_{\text{prior}}^2(\mathbf{y}; \psi)\})$, where $\boldsymbol{\sigma}_{\text{prior}}$ is estimated by the prior encoder $\psi$ with input $\mathbf{y}$. It appears that $\boldsymbol{\sigma}_{\text{prior}}$ follows the level variation of the speech waveform (in SE) and preserves the structure of the original image (in IR). This indicates that an informative prior approximating the data distribution has been obtained, leading to improved efficiency of the diffusion process.

- Experiments demonstrate that the proposed paradigm is quite general and can benefit both training and sampling of DDPMs, achieving considerable improvements with lightweight encoders in high quality signal restoration tasks of various modalities including images and audio.

## 2. Background on DDPMs

### 2.1. Forward Process

DDPMs (Ho et al., 2020; Sohl-Dickstein et al., 2015) slowly corrupt the training data using Gaussian noise in the forward

process. Let $q_{\text{data}}(\mathbf{x}_0)$ be the data density of the original data $\mathbf{x}_0$. The forward process is a fixed Markov Chain that sequentially corrupts the data $\mathbf{x}_0 \sim q_{\text{data}}(\mathbf{x}_0)$ in $T$ diffusion steps, by injecting Gaussian noise according to a variance schedule $\{\beta_t\}_{t=1}^{T} \in (0, 1)$:

$$q(\mathbf{x}_{1:T}|\mathbf{x}_0) \coloneqq \prod_{t=1}^{T} q(\mathbf{x}_t|\mathbf{x}_{t-1}), \qquad (1)$$

where $q(\mathbf{x}_t|\mathbf{x}_{t-1}) \coloneqq \mathcal{N}(\mathbf{x}_t; \sqrt{1-\beta_t}\mathbf{x}_{t-1}, \beta_t\mathbf{I})$ is the transition probability at step $t$. It allows the direct sampling of $\mathbf{x}_t$ according to $q(\mathbf{x}_t|\mathbf{x}_0) = \mathcal{N}(\mathbf{x}_t; \sqrt{\bar{\alpha}_t}\mathbf{x}_0, \sqrt{1-\bar{\alpha}_t}\mathbf{I})$, where $\bar{\alpha}_t \coloneqq \prod_{i=1}^{t} \alpha_i$ with $\alpha_t \coloneqq 1-\beta_t$; i.e., we can sample $\mathbf{x}_t = \sqrt{\bar{\alpha}_t}\mathbf{x}_0 + \sqrt{1-\bar{\alpha}_t}\boldsymbol{\epsilon}$, where $\boldsymbol{\epsilon} \sim \mathcal{N}(\mathbf{0}, \mathbf{I})$. A notable assumption is that with a carefully designed variance schedule $\beta_t$ and large enough $T$, such that $\bar{\alpha}_T$ is sufficiently small, $q(\mathbf{x}_T|\mathbf{x}_0)$ converges to $\mathcal{N}(\mathbf{x}_T; \mathbf{0}, \mathbf{I})$ so that the distribution of $\mathbf{x}_T$ is well approximated by the standard Gaussian.

## 2.2. Reverse Process

One can generate new data samples from $q_{\text{data}}(\mathbf{x}_0)$ by reversing the predefined forward process utilizing the same functional form. That is, we can progressively transform a noise $\mathbf{x}_T \sim p(\mathbf{x}_T) = \mathcal{N}(\mathbf{x}_T; \mathbf{0}, \mathbf{I})$ back into the data by approximating the reverse of the forward transition probability. This process is defined by the joint distribution $p_\theta(\mathbf{x}_{0:T})$ of a Markov Chain with learned transitions:

$$p_\theta(\mathbf{x}_{0:T}) \coloneqq p(\mathbf{x}_T) \prod_{t=1}^{T} p_\theta(\mathbf{x}_{t-1}|\mathbf{x}_t), \qquad (2)$$

where $p_\theta(\mathbf{x}_{t-1}|\mathbf{x}_t) \coloneqq \mathcal{N}(\mathbf{x}_{t-1}; \boldsymbol{\mu}_\theta(\mathbf{x}_t, t), \boldsymbol{\Sigma}_\theta(\mathbf{x}_t, t))$ is the reverse transition probability parameterized by a network $\theta$.

## 2.3. DDPM Learning Framework

Ideally, we would train the model $\theta$ with a maximum likelihood objective such that the probability assigned by the model $p_\theta(\mathbf{x}_0)$ to each training example is as large as possible, which is unfortunately intractable (Croitoru et al., 2023). To circumvent such difficulty, DDPMs (Ho et al., 2020) instead maximize an ELBO of the data log-likelihood, by introducing a sequence of hidden variables $\mathbf{x}_{1:T}$ and the approximate variational distribution $q(\mathbf{x}_{1:T}|\mathbf{x}_0)$:

$$\log p(\mathbf{x}_0) \geq \mathbb{E}_{q(\mathbf{x}_{1:T}|\mathbf{x}_0)}\left[\log \frac{p_\theta(\mathbf{x}_{0:T})}{q(\mathbf{x}_{1:T}|\mathbf{x}_0)}\right]. \qquad (3)$$

With the above parametric modeling of the forward and reverse processes, the ELBO (3) suggests training the network $\theta$ such that, at each time step $t$, $p_\theta(\mathbf{x}_{t-1}|\mathbf{x}_t)$ is as close as possible to the true forward process posterior conditioned on $\mathbf{x}_0$ (Luo, 2022; Croitoru et al., 2023), i.e., $q(\mathbf{x}_{t-1}|\mathbf{x}_t, \mathbf{x}_0) = \mathcal{N}(\mathbf{x}_{t-1}; \tilde{\boldsymbol{\mu}}_t(\mathbf{x}_t, \mathbf{x}_0), \tilde{\beta}_t\mathbf{I})$, where $\tilde{\beta}_t \coloneqq \frac{1-\bar{\alpha}_{t-1}}{1-\bar{\alpha}_t}\beta_t$ and $\tilde{\boldsymbol{\mu}}_t(\mathbf{x}_t, \mathbf{x}_0) \coloneqq \frac{\sqrt{\bar{\alpha}_{t-1}}\beta_t}{1-\bar{\alpha}_t}\mathbf{x}_0 + \frac{\sqrt{\alpha_t}(1-\bar{\alpha}_{t-1})}{1-\bar{\alpha}_t}\mathbf{x}_t$.

Based on using a fixed covariance $\boldsymbol{\Sigma}_\theta(\mathbf{x}_t, t) = \sigma_t^2\mathbf{I}$ (e.g., $\sigma_t^2 = \tilde{\beta}_t$) as in Ho et al. (2020), maximizing (3) corresponds to training a network $\boldsymbol{\mu}_\theta(\mathbf{x}_t, t)$ that predicts $\tilde{\boldsymbol{\mu}}_t(\mathbf{x}_t, \mathbf{x}_0)$. Alternatively, Ho et al. (2020) suggested the following reparameterization to rewrite the mean as a function of noise:

$$\boldsymbol{\mu}_\theta(\mathbf{x}_t, t) = \frac{1}{\sqrt{\alpha_t}}\left(\mathbf{x}_t - \frac{\beta_t}{\sqrt{1-\bar{\alpha}_t}}\boldsymbol{\epsilon}_\theta(\mathbf{x}_t, t)\right). \qquad (4)$$

They train a network $\boldsymbol{\epsilon}_\theta(\mathbf{x}_t, t)$ to predict the real noise $\boldsymbol{\epsilon} \sim \mathcal{N}(\mathbf{0}, \mathbf{I})$ and use (4) to compute the mean. Practically it is carried out by minimizing a simplified training objective:

$$\mathcal{L}_{\text{simple}}(\theta) \coloneqq \mathbb{E}_{\mathbf{x}_0, \boldsymbol{\epsilon}, t}\left[\|\boldsymbol{\epsilon} - \boldsymbol{\epsilon}_\theta(\mathbf{x}_t, t)\|^2\right], \qquad (5)$$

which measures, for a random time step $t \sim \mathcal{U}(\{1, \ldots, T\})$, the distance between the actual noise and estimated noise.

## 2.4. Signal Restoration by Conditional DDPMs

Signal restoration is concerned with recovering the original signals from their degraded observations, which are of paramount importance in reality but remaining challenging, as noises are ubiquitous and may be strong enough to cause significant degradation of the signal quality. Recently, adoption of deep generative models (Kingma & Welling, 2014; Goodfellow et al., 2014; Ho et al., 2020) for signal restoration tasks has considerably increased due to their remarkable capabilities of generating missing components in the data, with conditional DDPMs (Croitoru et al., 2023; Cao et al., 2024) demonstrating substantial promise. More formally, let $\mathbf{y}$ denote the degraded observation of the clean signal $\mathbf{x}_0$. Recovering $\mathbf{x}_0$ given $\mathbf{y}$ by a model $\theta$ can be cast as maximizing the conditional likelihood of data $p_\theta(\mathbf{x}_0|\mathbf{y})$. The problem is in general intractable, but can be approximated by using a DDPM conditioned on $\mathbf{y}$. The main idea is to learn a diffusion model $\theta$ with $\mathbf{y}$ provided as a conditioner in the reverse process (2):

$$p_\theta(\mathbf{x}_{0:T}|\mathbf{y}) \coloneqq p(\mathbf{x}_T) \prod_{t=1}^{T} p_\theta(\mathbf{x}_{t-1}|\mathbf{x}_t, \mathbf{y}), \qquad (6)$$

where $p_\theta(\mathbf{x}_{t-1}|\mathbf{x}_t, \mathbf{y}) \coloneqq \mathcal{N}(\mathbf{x}_{t-1}; \boldsymbol{\mu}_\theta(\mathbf{x}_t, \mathbf{y}, t), \sigma_t^2\mathbf{I})$ assuming a fixed covariance. In practice, a noise estimator network $\boldsymbol{\epsilon}_\theta(\mathbf{x}_t, \mathbf{y}, t)$ is adopted to predict the mean, following the practice in Ho et al. (2020).

# 3. Proposed Method: Integrating DDPM and VAE for Learnable Diffusion Prior

We start with the conditional VAE (Sohn et al., 2015) formulation to maximize the conditional data log-likelihood, $\log p(\mathbf{x}_0|\mathbf{y}) = \log \int p(\mathbf{x}_0, \boldsymbol{\epsilon}|\mathbf{y})d\boldsymbol{\epsilon}$, where $\boldsymbol{\epsilon}$ is an introduced latent variable. To avoid intractable integral, in VAEs an ELBO is utilized as the surrogate objective by introducing

an approximate posterior $q(\boldsymbol{\epsilon}|\mathbf{x}_0, \mathbf{y})$ (Harvey et al., 2022):

$$
\log p(\mathbf{x}_0|\mathbf{y}) \geq \underbrace{\mathbb{E}_{q(\boldsymbol{\epsilon}|\mathbf{x}_0,\mathbf{y})}\left[\log p(\mathbf{x}_0|\mathbf{y}, \boldsymbol{\epsilon})\right]}_{\text{reconstruction term}}
$$
$$
- \underbrace{D_{\mathrm{KL}}\left(q(\boldsymbol{\epsilon}|\mathbf{x}_0, \mathbf{y})||p(\boldsymbol{\epsilon}|\mathbf{y})\right)}_{\text{prior matching term}}. \qquad (7)
$$

The *reconstruction* and *prior matching* terms are typically realized by an encoder-decoder architecture with $\boldsymbol{\epsilon}$ being the bottleneck representation sampled from the latent distribution. VAEs generally benefit from learnable latent spaces for good modeling efficiency. However, their generative performance often lags behind DDPMs that employ an iterative, more sophisticated decoding (reconstruction) process.

In this work, our aim is to embrace the best of both worlds, i.e., *remarkable generative power (DDPM) and modeling efficiency (VAE) to achieve improved output signal quality and training/sampling efficiency simultaneously.*

**Proposition 3.1** (Incorporation of diffusion process into VAE). *By introducing a sequence of hidden variables $\mathbf{x}_{1:T}$, under the setup of conditional diffusion models where the Markov Chain assumption is employed on the forward process $q(\mathbf{x}_{1:T}|\mathbf{x}_0) := \prod_{t=1}^{T} q(\mathbf{x}_t|\mathbf{x}_{t-1})$ and the reverse process $p_\theta(\mathbf{x}_{0:T}|\mathbf{y}) := p(\mathbf{x}_T)\prod_{t=1}^{T} p_\theta(\mathbf{x}_{t-1}|\mathbf{x}_t, \mathbf{y})$ parameterized by a DDPM $\theta$, and assuming that $\mathbf{x}_T = \boldsymbol{\epsilon}$ (i.e., the latent noise of DDPM samples from the VAE latent distribution), we have the lower bound on $\log p(\mathbf{x}_0|\mathbf{y}, \boldsymbol{\epsilon})$ in the reconstruction term of the VAE (7) as:*

$$
\log p(\mathbf{x}_0|\mathbf{y}, \boldsymbol{\epsilon}) \geq \mathbb{E}_{q(\mathbf{x}_{1:T}|\mathbf{x}_0)}\left[\log \frac{p_\theta(\mathbf{x}_{0:T}|\mathbf{y})}{q(\mathbf{x}_{1:T}|\mathbf{x}_0)}\right], \qquad (8)
$$

*which is the ELBO of the conditional DDPM (i.e., the conditional version of (3)).*

The proof (based on Markov Chain property) is provided in Appendix A. Note that the assumption $\mathbf{x}_T = \boldsymbol{\epsilon}$ follows the standard DDPM to sample $\mathbf{x}_T$ from the distribution of the prior noise $\boldsymbol{\epsilon}$, practically achieved by using a large enough $T$ and a carefully designed variance schedule $\{\beta_t\}_{t=1}^{T}$. In our case, we adopt the same assumption to enable sampling $\mathbf{x}_T$ from the latent space of VAE. We interpret Proposition 3.1 as a seamless integration of DDPM into the VAE framework to achieve improved generative (decoding) capabilities.

Having incorporated the DDPM as the decoder, we now discuss the encoding part, i.e., the prior matching term in (7). A straightforward design could be using a network $\psi$ (*Prior Net*) to parameterize the prior distribution as $p_\psi(\boldsymbol{\epsilon}|\mathbf{y})$, while assuming the posterior to be a fixed form of distribution like the standard Gaussian. However, this may in turn discard any useful information inherent between $\mathbf{x}_0$ and $\mathbf{y}$. To take advantage of the adequate correlation present in signal restoration settings, we propose to also parameterize

the posterior distribution with another network $\phi$ (*Posterior Net*), to incorporate richer information about the target signal distribution into the learning of the prior. Together with (8), we introduce the **new lower bound** of the conditional data log-likelihood:

$$
\log p(\mathbf{x}_0|\mathbf{y}) \geq \mathbb{E}_{q_\phi(\boldsymbol{\epsilon}|\mathbf{x}_0,\mathbf{y})}\Bigg[\underbrace{\mathbb{E}_{q(\mathbf{x}_{1:T}|\mathbf{x}_0)}\left[\log \frac{p_\theta(\mathbf{x}_{0:T}|\mathbf{y})}{q(\mathbf{x}_{1:T}|\mathbf{x}_0)}\right]}_{\text{conditional DDPM}}\Bigg]
$$
$$
- D_{\mathrm{KL}}\big(\underbrace{q_\phi(\boldsymbol{\epsilon}|\mathbf{x}_0, \mathbf{y})}_{\text{Posterior Net}} || \underbrace{p_\psi(\boldsymbol{\epsilon}|\mathbf{y})}_{\text{Prior Net}}\big). \qquad (9)
$$

Based on the assumption $\mathbf{x}_T = \boldsymbol{\epsilon}$, the conditional DDPM samples the latent noise $\mathbf{x}_T$ from the distribution of $\boldsymbol{\epsilon}$ which is jointly estimated by the two encoders, $\phi$ and $\psi$. The two-encoder design is inspired by Kohl et al. (2018) for image segmentation with traditional U-Nets. Here, we adopt the idea in the context of DDPM for signal restoration, which is effective as it incorporates posterior information exploiting the correlation between clean and degraded signals. Based on (9), we introduce the training objective of RestoreGrad:

**Proposition 3.2** (RestoreGrad). *Assume the prior and posterior distributions are both zero-mean Gaussian, parameterized as $p_\psi(\boldsymbol{\epsilon}|\mathbf{y}) = \mathcal{N}(\boldsymbol{\epsilon}; \mathbf{0}, \boldsymbol{\Sigma}_{prior}(\mathbf{y}; \psi))$ and $q_\phi(\boldsymbol{\epsilon}|\mathbf{x}_0, \mathbf{y}) = \mathcal{N}(\boldsymbol{\epsilon}; \mathbf{0}, \boldsymbol{\Sigma}_{post}(\mathbf{x}_0, \mathbf{y}; \phi))$, respectively, where the covariances are estimated by the Prior Net $\psi$ (taking $\mathbf{y}$ as input) and Posterior Net $\phi$ (taking both $\mathbf{x}_0$ and $\mathbf{y}$ as input). Let us simply use $\boldsymbol{\Sigma}_{prior}$ and $\boldsymbol{\Sigma}_{post}$ hereafter to refer to $\boldsymbol{\Sigma}_{prior}(\mathbf{y}; \psi)$ and $\boldsymbol{\Sigma}_{post}(\mathbf{x}_0, \mathbf{y}; \phi)$ for concise notation. Then, with the direct sampling property in the forward path $\mathbf{x}_t = \sqrt{\bar{\alpha}_t}\mathbf{x}_0 + \sqrt{1-\bar{\alpha}_t}\boldsymbol{\epsilon}$ at arbitrary timestep $t$ where $\boldsymbol{\epsilon} \sim q_\phi(\boldsymbol{\epsilon}|\mathbf{x}_0, \mathbf{y})$, and assuming the reverse process has the same covariance as the true forward process posterior conditioned on $\mathbf{x}_0$, by utilizing the conditional DDPM $\boldsymbol{\epsilon}_\theta(\mathbf{x}_t, \mathbf{y}, t)$ as the noise estimator of the true noise $\boldsymbol{\epsilon}$, we have the modified ELBO, $-\mathcal{L}(\theta, \phi, \psi)$, associated with (9):*

$$
\mathcal{L}(\theta, \phi, \psi) = \underbrace{\frac{\bar{\alpha}_T}{2}\mathbb{E}_{\mathbf{x}_0}\|\mathbf{x}_0\|^2_{\boldsymbol{\Sigma}_{post}^{-1}} + \frac{1}{2}\log|\boldsymbol{\Sigma}_{post}|}_{\text{Latent Regularization (LR) terms}}
$$
$$
+ \underbrace{\sum_{t=1}^{T} \gamma_t \mathbb{E}_{(\mathbf{x}_0, \mathbf{y}), \boldsymbol{\epsilon} \sim \mathcal{N}(\mathbf{0}, \boldsymbol{\Sigma}_{post})}\|\boldsymbol{\epsilon} - \boldsymbol{\epsilon}_\theta(\mathbf{x}_t, \mathbf{y}, t)\|^2_{\boldsymbol{\Sigma}_{post}^{-1}}}_{\text{Denoising Matching (DM) terms}}
$$
$$
+ \underbrace{\frac{1}{2}\Big(\log\frac{|\boldsymbol{\Sigma}_{prior}|}{|\boldsymbol{\Sigma}_{post}|} + tr(\boldsymbol{\Sigma}_{prior}^{-1}\boldsymbol{\Sigma}_{post})\Big)}_{\text{Prior Matching (PM) terms}} + C,
$$

$$(10)$$

*where $\gamma_t = \begin{cases} \frac{\beta_t^2}{2\sigma_t^2\alpha_t(1-\bar{\alpha}_t)}, & t > 1 \\ \frac{1}{2\alpha_1}, & t = 1 \end{cases}$ are weighting factors, $\|\mathbf{x}\|^2_{\boldsymbol{\Sigma}^{-1}} = \mathbf{x}^T\boldsymbol{\Sigma}^{-1}\mathbf{x}$, $\sigma_t^2 = \frac{1-\bar{\alpha}_{t-1}}{1-\bar{\alpha}_t}\beta_t$ and $C$ is some constant not depending on learnable parameters $\theta$, $\phi$, $\psi$.*

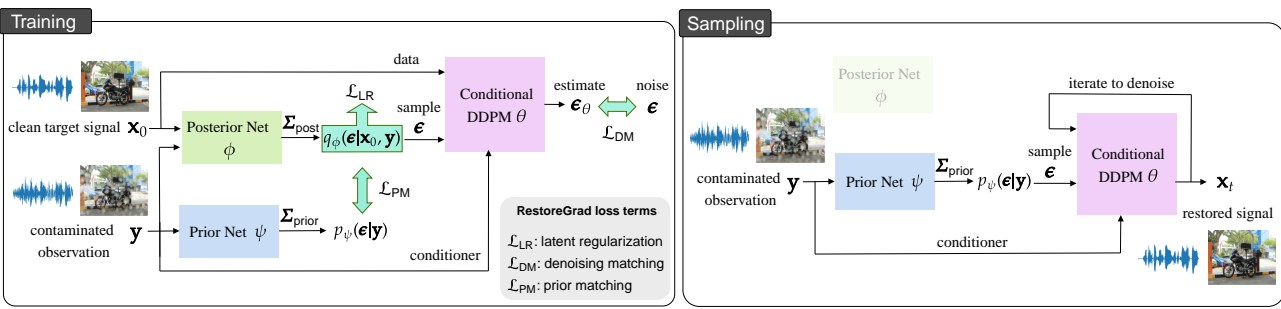

*Figure 4.* Proposed RestoreGrad. During training, the conditional DDPM $\theta$, Prior Net $\psi$, and Posterior Net $\phi$ are jointly optimized by (11). During inference, the DDPM $\theta$ samples the latent noise $\epsilon$ from the jointly learned prior distribution to synthesize the clean signal. (Summary of the algorithm details is presented in Appendix C.1.)

The derivation (see Appendix B) is based on combining VAE and the results in Lee et al. (2022). *Notably, we join the conditional DDPM with the posterior/prior encoders and optimize all modules at once, by connecting the DDPM prior space with the latent space estimated by the encoders.* To this end, the sampling $\epsilon \sim q_\phi(\epsilon|\mathbf{x}_0, \mathbf{y})$ is performed by the standard reparameterization trick as in VAEs, unlocking end-to-end training via gradient descent on the loss terms:

- *Latent Regularization (LR)* terms: to help learn a reasonable latent space; e.g., minimizing $\log|\boldsymbol{\Sigma}_{\text{post}}|$ avoids $\boldsymbol{\Sigma}_{\text{post}}$ from becoming arbitrary large due to the presence of its inverse in the weighted norms.

- *Denoising Matching (DM)* terms: responsible for training the DDPM to predict the true noise.

- *Prior Matching (PM)* terms: to shape a desirable latent space by aligning the prior and posterior distributions. Note that we model the distributions as zero-mean, based on that signals can be properly normalized.

### 3.1. Training of RestoreGrad

With the the conditional DDPM $\theta$, Prior Net $\psi$, and Posterior Net $\phi$ defined in Proposition 3.2, optimization can be performed to learn the model parameters of $\theta, \psi, \phi$ based on the modified ELBO. The RestoreGrad framework jointly trains the three neural network modules by minimizing (10) as depicted in Figure 4. Following existing DDPM literature, we approximate the objective by dropping the weighting constant $\gamma_t$ of the DM terms, leading to the simplified loss:

$$\min_{\theta,\phi,\psi} \ \eta\big(\underbrace{\bar{\alpha}_T\|\mathbf{x}_0\|^2_{\boldsymbol{\Sigma}^{-1}_{\text{post}}} + \log|\boldsymbol{\Sigma}_{\text{post}}|}_{\mathcal{L}_{\text{LR}}}\big) + \underbrace{\|\epsilon - \boldsymbol{\epsilon}_\theta(\mathbf{x}_t, \mathbf{y}, t)\|^2_{\boldsymbol{\Sigma}^{-1}_{\text{post}}}}_{\mathcal{L}_{\text{DM}}}$$

$$+\lambda\underbrace{\big(\log\frac{|\boldsymbol{\Sigma}_{\text{prior}}|}{|\boldsymbol{\Sigma}_{\text{post}}|} + \text{tr}(\boldsymbol{\Sigma}^{-1}_{\text{prior}}\boldsymbol{\Sigma}_{\text{post}})\big)}_{\mathcal{L}_{\text{PM}}},$$

(11)

where we approximate the expectations by randomly sampling $(\mathbf{x}_0, \mathbf{y}) \sim q_{\text{data}}(\mathbf{x}_0, \mathbf{y})$ and $\epsilon \sim \mathcal{N}(\mathbf{0}, \boldsymbol{\Sigma}_{\text{post}})$, and the

summation over $t$ by sampling $t \sim \mathcal{U}(\{1, \dots, T\})$ (exploiting the independency due to Markov assumption (Nichol & Dhariwal, 2021)) in each training iteration. We have also introduced $\eta > 0$ for the LR terms and $\lambda > 0$ for PM terms, to exert flexible control of the learned latent space.

### 3.2. Sampling of RestoreGrad

In applications that RestoreGrad is mainly concerned with, the target signal $\mathbf{x}_0$ is not available in inference time. As in Figure 4, the conditional DDPM then samples $\epsilon \sim p_\psi(\epsilon|\mathbf{y}) = \mathcal{N}(\mathbf{0}, \boldsymbol{\Sigma}_{\text{prior}})$ from the Prior Net instead; the Posterior Net is no longer needed.

### 3.3. The Role of Posterior Information

In the training stage of RestoreGrad, the latent code $\epsilon$ samples from the posterior $q_\phi(\epsilon|\mathbf{x}_0, \mathbf{y})$ which exploits both the ground truth signal $\mathbf{x}_0$ and conditioner $\mathbf{y}$. It is thus more advantageous than existing works on adaptive priors (e.g., PriorGrad (Lee et al., 2022)) that only utilize the conditioner $\mathbf{y}$. To observe the benefits brought by the posterior information, we can make comparison with a variant of RestoreGrad where the Posterior Net is excluded in training:

$$\min_{\theta,\psi} \ \eta\big(\bar{\alpha}_T\|\mathbf{x}_0\|^2_{\boldsymbol{\Sigma}^{-1}_{\text{prior}}} + \log|\boldsymbol{\Sigma}_{\text{prior}}|\big) + \|\epsilon - \boldsymbol{\epsilon}_\theta(\mathbf{x}_t, \mathbf{y}, t)\|^2_{\boldsymbol{\Sigma}^{-1}_{\text{prior}}},$$

(12)

which basically removes the Posterior Net $\phi$ and only trains the Prior Net $\psi$ and DDPM $\theta$. Interestingly, our experimental results show that RestoreGrad indeed performs better with the Posterior Net than without it in model training.

## 4. Experiments

### 4.1. Application to Speech Enhancement (SE)

#### 4.1.1. EXPERIMENTAL SETUP

**Dataset:** We validate performance on the benchmark SE dataset *VoiceBank+DEMAND* (Valentini-Botinhao et al., 2016), consisting of clean speech clips collected from the *VoiceBank* corpus (Veaux et al., 2013), mixed with ten types

of noise profiles from the *DEMAND* database (Thiemann et al., 2013). Specifically, the training utterances from Voice-Bank are artificially contaminated with the noise samples from DEMAND at 0, 5, 10, and 15 dB signal-to-noise ratio (SNR) levels, amounting to 11,572 utterances. The testing utterances are mixed with different noise samples at 2.5, 7.5, 12.5, and 17.5 dB SNR levels, amounting to 824 utterances.

**Evaluation Metrics:** We consider: **PESQ:** Perceptual Evaluation of Speech Quality (ITU-T Rec. P.862.2, 2005). **SI-SNR:** Scale-Invariant SNR (Le Roux et al., 2019). **SSNR:** Segmental SNR (Hu & Loizou, 2007). **CSIG, CBAK, COVL:** Mean-opinion-score predictors of signal distortion, background-noise intrusiveness, and overall signal quality, respectively (Hu & Loizou, 2007).

**Models:** The following models are compared:

- **Baseline DDPM**: We adopt the *CDiffuSE (Base)* model from Lu et al. (2022), which is based on DiffWave (Kong et al., 2021) with 4.28M learnable parameters.

- **PriorGrad**: We implement the PriorGrad (Lee et al., 2022) on top of CDiffuSE by changing the prior distribution from $\mathcal{N}(\mathbf{0}, \mathbf{I})$ to $\mathcal{N}(\mathbf{0}, \mathbf{\Sigma}_\mathrm{y})$, where $\mathbf{\Sigma}_\mathrm{y}$ is the covariance of the data-dependent prior computed based on the conditioner $\mathbf{y}$, using the rule-based estimation approach for the application to vocoder in Lee et al. (2022).

- **RestoreGrad**: We incorporate Prior Net and Posterior Net on top of CDiffuSE. Both modules adopt the ResNet-20 architect (He et al., 2016), suitably modified to 1-D convolutions for waveform processing, each with only 93K learnable parameters (only 2% of the CDiffuSE model).

**Configurations:** We adopted the basic configurations same as in Lu et al. (2022). The waveforms were processed at 16kHz sampling rate. The number of forward diffusion steps was $T = 50$. The variance schedule was $\beta_t \in [10^{-4}, 0.035]$, linearly spaced. The batch size was 16. The fast sampling scheme in Kong et al. (2021) was used in the reverse process with 6 steps to reduce inference complexity, with the 6-step inference variance schedule $\beta_t^{\mathrm{infer}} = [10^{-4}, 10^{-3}, 0.01, 0.05, 0.2, 0.35]$. Adam optimizer (Kingma & Ba, 2014) was utilized with a learning rate of $2 \times 10^{-4}$. We set $\eta = 0.1$ and $\lambda = 0.5$ for (11).

### 4.1.2. RESULTS

**Improved Model Convergence:** As shown in Figure 2 (test set performance), RestoreGrad shows better convergence behavior over PriorGrad (using handcrafted prior) and CDiffuSE (using standard Gaussian prior). For example, *PriorGrad reaches 2.4 in PESQ at 96 epochs, whereas RestoreGrad reaches it in (roughly) 10 epochs, indicating a 10× speed-up.* The results suggest that jointly learning the prior distribution can be beneficial for conditional DDPMs.

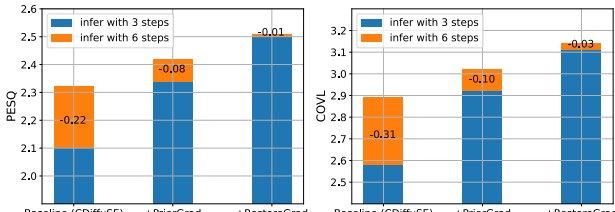

*Figure 5.* Robustness to the reduction in reverse sampling time steps for inference.

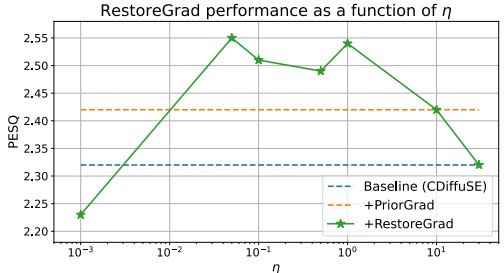

*Figure 6.* Effect of latent regularization weight $\eta$ for $\mathcal{L}_{\mathrm{LR}}$ on SE.

**Robustness to Reduced Reverse Steps in Inference:** RestoreGrad can potentially reduce the inference complexity. In Figure 5, we show how the trained diffusion models withstand the reduction in the number of inference steps. In each model, we trained the network for 96 epochs and then inferenced with 3 reverse steps to compare with the originally adopted 6-step scheme in Lu et al. (2022). The noise schedule for the 3-step scheme was $\beta_t^{\mathrm{infer}} = [0.05, 0.2, 0.35]$, a subset of the 6-step schedule that resulted in best performance. We can see that the baseline DDPM is most sensitive to the step reduction, while PriorGrad shows certain resistance as leveraging a closer-to-data prior distribution. *Finally, RestoreGrad barely degrades with reduced sampling steps, echoing that a better prior has been obtained as it recovers higher fidelity signal even in fewer reverse steps.*

**Effect of $\eta$:** An important factor in our prior learning scheme is the regularization weight $\eta$ for $\mathcal{L}_{\mathrm{LR}}$ of the training loss. An appropriate value of $\eta$ should be large enough to properly regularize the learned latent space for avoiding instability, while not adversely affecting signal reconstruction performance. It is thus interesting to see how the performance varies with the choice of $\eta$. *Empirically, we found the overall SE performance not to be very sensitive to the value of $\eta$ across a wide range,* as shown in Figure 6: Roughly in the range of $[10^{-2}, 10]$ of the $\eta$ value we see that RestoreGrad gives better results over both PriorGrad and CDiffuSE.

**Comparison to Fully-Trained CDiffuSE:** We present in Table 1 more detailed comparison of RestoreGrad with the baseline CDiffuSE. Here, the scores of CDiffuSE were directly taken from the results reported in Lu et al. (2022) where the model has been fully trained for 445 epochs. For

*Table 1.* Comparison with the fully-trained CDiffuSE model performance reported in Lu et al. (2022).

| Methods | # train epochs | # infer steps | PESQ↑ | CSIG↑ | CBAK↑ | COVL↑ | SI-SNR↑ |
|---|---|---|---|---|---|---|---|
| CDiffuSE | 445 | 6 | 2.44 | 3.66 | 2.83 | 3.03 | - |
| + PriorGrad | 96 | 6 | 2.42±3e-3 | 3.67±2e-3 | 2.93±1e-3 | 3.03±2e-3 | 14.21±2e-3 |
| + RestoreGrad (ours) | 96 | 6 | **2.51**±6e-4 | **3.80**±4e-4 | **3.00**±3e-4 | **3.14**±5e-4 | **14.74**±3e-4 |
| | | 3 | 2.50±3e-4 | 3.75±2e-4 | 2.99±2e-4 | 3.11±3e-4 | 14.65±2e-4 |

*Bold text for best and underlined text for second best values.

*Table 2.* SE comparison of RestoreGrad models using encoder modules of different sizes and the corresponding latency and GPU memory usage (measured on one NVIDIA Tesla V100 GPU) presented as the ratio of encoder to DDPM.

| Encoder size | PESQ↑ | COVL↑ | SSNR↑ | SI-SNR↑ | Proc. Time | Memory |
|---|---|---|---|---|---|---|
| Tiny (24K) | 2.48 | 3.11 | 5.10 | 13.74 | 1.9% | 6.5% |
| Base (93K) | 2.51 | 3.14 | 5.92 | 14.74 | 2.2% | 10.3% |
| Large (370K) | 2.54 | 3.16 | 6.15 | 15.01 | 2.6% | 18.2% |

PriorGrad and RestoreGrad we report the mean±std computed based on results of 10 independent samplings. *We can see that with RestoreGrad applied, the SE model can achieve better performance over the baseline CDiffuSE by only training for 96 epochs (4.6 times lesser) in all the metrics.* In addition, halving the number of reverse steps in inference time still maintains better performance than the fully-trained CDiffuSE and also the PriorGrad.

**Signal Quality and Encoder Complexity Trade-Offs:** We further present results using three different model sizes (24K, 93K, 370K) for the Prior and Posterior Nets (encoders) in Table 2, along with latency and GPU memory usage (presented as the ratio of encoder to DDPM). The results clearly show that the restored speech quality improves with increasing encoder size. *This indicates that there is a trade-off between the restoration signal quality and encoder model complexity.* Notably, *the latency and memory usage of the encoder modules are relatively small compared to the DDPM decoding (<2.6% latency and <18.2% memory usage of the DDPM processing),* suggesting that RestoreGrad is capable of achieving improved performance without incurring considerable increase in complexity compared to the adopted DDPM model.

**Posterior Net Helps:** Finally, we validate the benefits brought by employing Posterior Net in the training phase by comparing with the RestoreGrad models trained without Posterior Net as (12) for some $\eta$. For fairness, all models were trained with 96 epochs, inferred with 6 steps. In Table 3, we observe that *RestoreGrad achieves better results with Posterior Net than without it, indicating the benefits of being informed of the target $\mathbf{x}_0$ by utilizing the Posterior Net.* We also observe that without regularizing the latent space (i.e., with $\eta = 0$) it could lead to training divergence.

*Table 3.* Performance of RestoreGrad models trained with and without using Posterior Net.

| SE models | PESQ↑ | COVL↑ | SSNR↑ | SI-SNR↑ |
|---|---|---|---|---|
| CDiffuSE (trained for 96 epochs) | 2.32 | 2.89 | 3.94 | 11.84 |
| + PriorGrad | 2.42 | 3.03 | 5.53 | 14.21 |
| + RestoreGrad | **2.51** | **3.14** | **5.92** | **14.74** |
| + RestoreGrad w/o Posterior Net ($\eta = 0$) | — | training | diverged | — |
| + RestoreGrad w/o Posterior Net ($\eta = 0.01$) | 2.47 | 3.08 | 4.96 | 11.22 |
| + RestoreGrad w/o Posterior Net ($\eta = 1$) | 2.48 | 3.12 | 5.11 | 13.29 |

*Best values in bold.

## 4.2. Application to Image Restoration (IR)

### 4.2.1. EXPERIMENTAL SETUP

**Dataset:** Following Özdenizci & Legenstein (2023), we consider the IR task of recovering clean images from their degraded versions contaminated by synthesized noises corresponding to different weather conditions. Two datasets are considered, where one is a weather-specific dataset called *RainDrop* (Qian et al., 2018) and the other is a multi-weather dataset named *AllWeather* (Valanarasu et al., 2022). The RainDrop dataset consists of images captured with raindrops on the camera sensor which obstruct the view. It has 861 training images and a test set of 58 images dedicated for quantitative evaluations. The AllWeather dataset is a curated training dataset from Valanarasu et al. (2022), which has 18,069 samples composed of subsets of training images from *Snow100K* (Liu et al., 2018), *Outdoor-Rain* (Li et al., 2019) and *RainDrop* (Qian et al., 2018), in order to create a balanced training set across three weather conditions.

**Evaluation Metrics:** Quantitative evaluations of restored images are performed via Peak Signal-to-Noise Ratio (**PSNR**) (Huynh-Thu & Ghanbari, 2008), Structural SIMilarity (**SSIM**) (Wang et al., 2004), Learned Perceptual Image Patch Similarity (**LPIPS**) (Zhang et al., 2018), and Fréchet Inception Distance (**FID**) (Heusel et al., 2017).

**Models:** The following IR models are compared:

- **Baseline DDPMs**: We consider the *RainDropDiff*$_{64}$ and *WeatherDiff*$_{64}$ in Özdenizci & Legenstein (2023) trained on the RainDrop and AllWeather datasets, respectively, as baseline DDPMs. Our work is based on the implementation provided by Özdenizci & Legenstein (2023).

- **RestoreGrad**: We incorporate the encoder modules, Prior Net and Posterior Net, on top of the baseline DDPMs. Both encoder modules adopt the ResNet-20 architect (He et al., 2016) with only 0.27M learnable parameters, significantly smaller (< 0.3%) than the baseline DDPM models.

**Configurations:** We used Adam optimizer with a learning rate of $2 \times 10^{-5}$. An exponential moving average with a weight of 0.999 was applied. We used $T = 1000$ and linear noise schedule $\beta_t \in [10^{-4}, 0.02]$, same as Özdenizci & Legenstein (2023). A batch size of 4 was used.

*Table 4.* Comparison with existing IR models. The multi-weather (MW) models were trained on the AllWeather training set (Valanarasu et al., 2022) and tested on three different weather types: Snow100K-L (Liu et al., 2018), Outdoor-Rain (Li et al., 2019), and RainDrop (Qian et al., 2018). Several weather-specific (WS) models that were trained on individual weather types are also presented for reference.

| Type | Methods | Snow100K-L | | Methods | Outdoor-Rain | | Methods | RainDrop | |
|---|---|---|---|---|---|---|---|---|---|
| | | PSNR ↑ | SSIM ↑ | | PSNR ↑ | SSIM ↑ | | PSNR ↑ | SSIM ↑ |
| WS | RESCAN (Li et al., 2018) | 26.08 | 0.8108 | HRGAN (Li et al., 2019) | 21.56 | 0.8550 | AttentiveGAN (Qian et al., 2018) | 31.59 | 0.9170 |
| | DesnowNet (Liu et al., 2018) | 27.17 | 0.8983 | PCNet (Jiang et al., 2021) | 26.19 | 0.9015 | RaindropAttn (Quan et al., 2019) | 31.44 | 0.9263 |
| | DDMSNet (Zhang et al., 2021) | 28.85 | 0.8772 | MPRNet (Zamir et al., 2021) | 28.03 | 0.9192 | IDT (Xiao et al., 2022) | 31.87 | 0.9313 |
| | SnowDiff (Özdenizci & Legenstein, 2023) | 30.43 | 0.9145 | RainHazeDiff (Özdenizci & Legenstein, 2023) | 28.38 | 0.9320 | RainDropDiff (Özdenizci & Legenstein, 2023) | 32.29 | 0.9422 |
| | DTPM (Ye et al., 2024) | 30.92 | 0.9174 | DTPM (Ye et al., 2024) | 30.99 | 0.9340 | DTPM (Ye et al., 2024) | **32.72** | **0.9440** |
| MW | All-in-One (Li et al., 2020) | 28.33 | 0.8820 | All-in-One (Li et al., 2020) | 24.71 | 0.8980 | All-in-One (Li et al., 2020) | 31.12 | 0.9268 |
| | TransWeather (Valanarasu et al., 2022) | 29.31 | 0.8879 | TransWeather (Valanarasu et al., 2022) | 28.83 | 0.9000 | TransWeather (Valanarasu et al., 2022) | 30.17 | 0.9157 |
| | WeatherDiff (Özdenizci & Legenstein, 2023) | 30.09 | 0.9041 | WeatherDiff (Özdenizci & Legenstein, 2023) | 29.64 | 0.9312 | WeatherDiff (Özdenizci & Legenstein, 2023) | 30.71 | 0.9312 |
| | + RestoreGrad (ours) | 30.82 | 0.9159 | + RestoreGrad (ours) | 30.83 | 0.9411 | + RestoreGrad (ours) | 31.78 | 0.9394 |
| | + RestoreGrad (ours) – trained longer | **31.16** | **0.9175** | + RestoreGrad (ours) – trained longer | 30.70 | **0.9418** | + RestoreGrad (ours) – trained longer | 32.26 | 0.9414 |

*Bold text for best and underlined text for second best values.

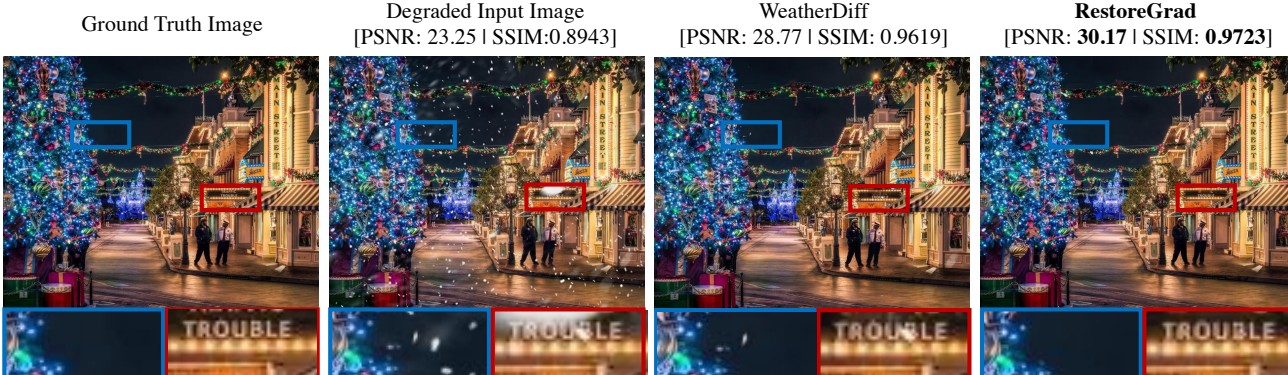

*Figure 7.* Image restoration examples using a test image taken from the Snow100K-L test set. We provide more examples, including other degradations (**desnowing**, **deraining**, **raindrop removal** and **deblurring**) in Appendix D.2.

### 4.2.2. RESULTS

**Model Convergence:** As presented in Figure 2 (test set performance), RestoreGrad demonstrates faster convergence and better restored image quality over the baseline DDPM (RainDropDiff). For example, *RainDropDiff reaches 0.143 in LPIPS at 9.2K epochs, while RestoreGrad reaches it in 1.8K epochs only, indicating a 5× speed-up due to the effectiveness of the joint prior learning scheme.*

**Comparison with Existing IR Models:** We compare our method with existing models of the multi-weather (MW) type in Table 4, including All-in-One (Li et al., 2020) and TransWeather (Valanarasu et al., 2022) in addition to WeatherDiff (Özdenizci & Legenstein, 2023). The models were all trained on the AllWeather dataset and the performance numbers were taken from Özdenizci & Legenstein (2023), where the WeatherDiff was trained for 1,775 epochs and inferenced with 25 steps. *Our RestoreGrad was trained for only 887 epochs (2× fewer) and inferenced with 10 steps to already achieve the best performance in the MW category.* In addition, when trained for more epochs (1,551

epochs, which is 1.14× fewer than WeatherDiff), Restore-Grad achieves further improvements as shown in the last row the table. On the other hand, our method, while trained on multi-weather data, achieves a comparable performance to the recently proposed Diffusion Texture Prior Model (DTPM) (Ye et al., 2024) individually trained on the three weather-specific (WS) datasets. *Notably, our method does not require a pretraining stage on a large dataset of high-quality images, unlike DTPM which is pretrained on 55,000 images samples.* This suggests the generality and effectiveness of our method to improve baseline DDPM models.

**IR Example:** Figure 7 presents examples of restored images by the models. It can be seen that RestoreGrad is able to better recover the original image, *especially in regions of the blue and red boxes where the baseline WeatherDiff fails to remove the snow obstructions.* The higher PSNR and SSIM scores of RestoreGrad also reflect the improvements.

**Other IR Tasks:** Our method demonstrates the advantages of learnable priors in image *deblurring* and *super-resolution* tasks, suggesting its *generality*. See Appendix D.2.

*Table 5.* Evaluation on realistic image datasets of the IR models trained on synthetic images of AllWeather training set.

| Methods | Gen. | RainDS-Real NIQE ↓ | Snow-Real NIQE ↓ |
|---|---|---|---|
| TransWeather (Valanarasu et al., 2022) | N | 4.005 | 3.161 |
| WeatherDiff (Özdenizci & Legenstein, 2023) | Y | 3.050 | **2.985** |
| + RestoreGrad (ours) | Y | **2.556** | 3.015 |

\*Bold text for best and underlined text for second best values. The column "Gen."

indicates if the model is generative (Y) or not (N).

*Table 6.* Evaluation of SE models on CHiME-3 test set, where the models were trained on VoiceBank+DEMAND training set.

| Methods | Gen. | PESQ↑ | CSIG↑ | CBAK↑ | COVL↑ | SI-SNR↑ |
|---|---|---|---|---|---|---|
| Unprocessed | - | 1.27 | 2.61 | 1.93 | 1.88 | 7.51 |
| Demucs (Defossez et al., 2020) | N | 1.38 | 2.50 | 2.08 | 1.88 | - |
| WaveCRN (Hsieh et al., 2020) | N | 1.43 | 2.53 | 2.03 | 1.91 | - |
| DOSE (Tai et al., 2023a) | Y | 1.52 | 2.71 | **2.15** | 2.06 | - |
| CDiffuSE (Lu et al., 2022) | Y | **1.55** | 2.87 | 2.09 | 2.15 | 7.67 |
| + RestoreGrad (ours) | Y | 1.54 | **2.88** | 2.14 | **2.16** | **8.45** |

\*Bold text for best and underlined text for second best values. The column "Gen."

indicates if the model is generative (Y) or not (N).

## 4.3. Generalization to Out-of-Distribution (OOD) and Realistic Data

We have so far evaluated the models on in-domain scenarios with synthetic noisy data where RestoreGrad has shown substantial improvements. A natural question is whether the demonstrated improvements have actually come at the expense of the model's generalizability to unseen or realistic data. To address the concern, we evaluate the IR models on two additional datasets, *RainDS-Real* from Quan et al. (2021) and *Snow100K-Real* from Liu et al. (2018) consisting of real-world images, using the reference-free Natural Image Quality Evaluator (NIQE) metric (Mittal et al., 2012) (a lower score indicates better quality). In Table 5 we see that RestoreGrad is able to perform on par with or better than WeatherDiff and the non-generative model of TransWeather. For OOD testing, we evaluate the SE models on the *CHiME-3* dataset (Barker et al., 2017) unseen during model training. Table 6 compares RestoreGrad with CDiffuSE also trained for 96 epochs, DOSE (Tai et al., 2023a), and two discriminative SE models, Demucs (Defossez et al., 2020) and WaveCRN (Hsieh et al., 2020). We can see that RestoreGrad is able to perform equally well as the CDiffuSE while outperforming DOSE and the non-generative SE models. The results show that RestoreGrad is capable of improving in-domain performance while maintaining desirable generalizability of generative models.

## 5. Related Work

**Diffusion Efficiency Improvements:** Das et al. (2023) utilized the shortest path between two Gaussians and Song et al. (2020) generalized DDPMs via a class of non-Markovian

diffusion processes to reduce the number of diffusion steps. Nichol & Dhariwal (2021) introduced a few simple modifications to improve the log-likelihood. Pandey et al. (2022; 2021) used DDPMs to refine VAE-generated samples. Rombach et al. (2022) performed the diffusion process in the lower dimensional latent space of an autoencoder to achieve high-resolution image synthesis, and Liu et al. (2023b) studied using such latent diffusion models for audio. Popov et al. (2021) explored using a text encoder to extract better representations for continuous-time diffusion-based text-to-speech generation. More recently, Nielsen et al. (2024) explored using a time-dependent image encoder to parameterize the mean of the diffusion process. Orthogonal to the above, PriorGrad (Lee et al., 2022) and follow-up work (Koizumi et al., 2022) studied utilizing informative prior extracted from the conditioner data for improving learning efficiency. *However, they become sub-optimal when the conditioner are degraded versions of the target data, posing challenges in applications like signal restoration tasks.*

**Diffusion-Based Signal Restoration:** Built on top of the diffusion models for audio generation, e.g., Kong et al. (2021); Chen et al. (2020); Leng et al. (2022), many SE models have been proposed. The pioneering work of Lu et al. (2022) introduced conditional DDPMs to the SE task and demonstrated the potential. Other works (Serrà et al., 2022; Welker et al., 2022; Richter et al., 2023; Yen et al., 2023; Lemercier et al., 2023; Tai et al., 2023a) have also attempted to improve SE by exploiting diffusion models. In the vision domain, diffusion models have demonstrated impressive performance for IR tasks (Li et al., 2023; Zhu et al., 2023; Huang et al., 2024; Luo et al., 2023; Xia et al., 2023; Fei et al., 2023; Hurault et al., 2022; Liu et al., 2023a; Chung et al., 2023b;a; Zhou et al., 2024; Xiao et al., 2024; Zheng et al., 2025; Liu et al., 2024; Ye et al., 2024), and a comprehensive view of recent advancements is provided by He et al. (2025). A notable IR work is by Özdenizci & Legenstein (2023) that achieved impressive performance on several benchmark datasets for restoring vision in adverse weather conditions. *Despite showing promising results, existing works have not fully exploited prior information about the data as they mostly settle on standard Gaussian priors.*

## 6. Conclusion

We investigated the potential of learning the prior distribution in tandem with the conditional DDPM for improved efficiency. We demonstrated the advantages of RestoreGrad that leverages learning-based priors, providing a more systematic way of estimating the prior than existing selections. A limitation of the current work is that it focuses on signal restoration applications, where we suitably assume a zero-mean Gaussian prior and only learn its covariance. In the future, it can be interesting to explore using a more generic prior form and extend the idea to other applications.

## Impact Statement

This paper presents work whose goal is to advance the field of Machine Learning. There are many potential societal consequences of our work, none of which we feel must be specifically highlighted here.

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

## A. Proof of Proposition 3.1

**Proposition 3.1** *(Incorporation of diffusion process into VAE). By introducing a sequence of hidden variables $\mathbf{x}_{1:T}$, under the setup of conditional diffusion models where the Markov Chain assumption is employed on the forward process $q(\mathbf{x}_{1:T}|\mathbf{x}_0) := \prod_{t=1}^{T} q(\mathbf{x}_t|\mathbf{x}_{t-1})$ and the reverse process $p_\theta(\mathbf{x}_{0:T}|\mathbf{y}) := p(\mathbf{x}_T) \prod_{t=1}^{T} p_\theta(\mathbf{x}_{t-1}|\mathbf{x}_t, \mathbf{y})$ parameterized by a DDPM $\theta$, and assuming that $\mathbf{x}_T = \boldsymbol{\epsilon}$ (i.e., the latent noise of DDPM samples from the VAE latent distribution), we have the lower bound on $\log p(\mathbf{x}_0|\mathbf{y}, \boldsymbol{\epsilon})$ in the reconstruction term of the VAE (7) as:*

$$\log p(\mathbf{x}_0|\mathbf{y}, \boldsymbol{\epsilon}) \geq \mathbb{E}_{q(\mathbf{x}_{1:T}|\mathbf{x}_0)} \left[ \log \frac{p_\theta(\mathbf{x}_{0:T}|\mathbf{y})}{q(\mathbf{x}_{1:T}|\mathbf{x}_0)} \right],$$

*which is the ELBO of the conditional DDPM (i.e., the conditional version of (3)).*

*Proof:*

We lower bound the conditional data log-likelihood $\log p(\mathbf{x}_0|\mathbf{y}, \boldsymbol{\epsilon})$ by utilizing a sequence of hidden latent representations $\mathbf{x}_{1:T}$ and the approximate variational distribution $q(\mathbf{x}_{1:T}|\mathbf{x}_0)$:

$$\begin{aligned}
\log p(\mathbf{x}_0|\mathbf{y}, \boldsymbol{\epsilon}) &= \log \int p_\theta(\mathbf{x}_{0:T}|\mathbf{y}, \boldsymbol{\epsilon}) d\mathbf{x}_{1:T} \\
&= \log \int \frac{p_\theta(\mathbf{x}_{0:T}|\mathbf{y}, \boldsymbol{\epsilon}) q(\mathbf{x}_{1:T}|\mathbf{x}_0)}{q(\mathbf{x}_{1:T}|\mathbf{x}_0)} d\mathbf{x}_{1:T} \\
&= \log \mathbb{E}_{q(\mathbf{x}_{1:T}|\mathbf{x}_0)} \left[ \frac{p_\theta(\mathbf{x}_{0:T}|\mathbf{y}, \boldsymbol{\epsilon})}{q(\mathbf{x}_{1:T}|\mathbf{x}_0)} \right] \\
&\geq \mathbb{E}_{q(\mathbf{x}_{1:T}|\mathbf{x}_0)} \left[ \log \frac{p_\theta(\mathbf{x}_{0:T}|\mathbf{y}, \boldsymbol{\epsilon})}{q(\mathbf{x}_{1:T}|\mathbf{x}_0)} \right],
\end{aligned} \tag{13}$$

where the inequality is obtained by applying Jensen's inequality.

The reverse diffusion process incorporating $\boldsymbol{\epsilon}$ is given as:

$$\begin{aligned}
p_\theta(\mathbf{x}_{0:T}|\mathbf{y}, \boldsymbol{\epsilon}) &:= p(\mathbf{x}_T) \prod_{t=1}^{T} p_\theta(\mathbf{x}_{t-1}|\mathbf{x}_t, \mathbf{y}, \boldsymbol{\epsilon}) \\
&= p(\mathbf{x}_T) \prod_{t=1}^{T} p_\theta(\mathbf{x}_{t-1}|\mathbf{x}_t, \mathbf{y}, \mathbf{x}_T) \\
&= p(\mathbf{x}_T) \prod_{t=1}^{T} p_\theta(\mathbf{x}_{t-1}|\mathbf{x}_t, \mathbf{y}) =: p_\theta(\mathbf{x}_{0:T}|\mathbf{y}),
\end{aligned} \tag{14}$$

by using the assumption that the diffusion prior samples from the VAE latent space, i.e., $\mathbf{x}_T = \boldsymbol{\epsilon}$, and utilizing the Markov Chain property on the reverse transition probability to remove the $\mathbf{x}_T$ as a condition. This indicates that:

$$\mathbb{E}_{q(\mathbf{x}_{1:T}|\mathbf{x}_0)} \left[ \log \frac{p_\theta(\mathbf{x}_{0:T}|\mathbf{y}, \boldsymbol{\epsilon})}{q(\mathbf{x}_{1:T}|\mathbf{x}_0)} \right] = \mathbb{E}_{q(\mathbf{x}_{1:T}|\mathbf{x}_0)} \left[ \log \frac{p_\theta(\mathbf{x}_{0:T}|\mathbf{y})}{q(\mathbf{x}_{1:T}|\mathbf{x}_0)} \right]. \tag{15}$$

Together, (13) and (15) lead to the result in Proposition 3.1.

## B. Derivation of Proposition 3.2

**Proposition 3.2** (RestoreGrad). *Assume the prior and posterior distributions are both zero-mean Gaussian, parameterized as $p_\psi(\boldsymbol{\epsilon}|\mathbf{y}) = \mathcal{N}(\boldsymbol{\epsilon}; \mathbf{0}, \boldsymbol{\Sigma}_{prior}(\mathbf{y}; \psi))$ and $q_\phi(\boldsymbol{\epsilon}|\mathbf{x}_0, \mathbf{y}) = \mathcal{N}(\boldsymbol{\epsilon}; \mathbf{0}, \boldsymbol{\Sigma}_{post}(\mathbf{x}_0, \mathbf{y}; \phi))$, respectively, where the covariances are estimated by the Prior Net $\psi$ (taking $\mathbf{y}$ as input) and Posterior Net $\phi$ (taking both $\mathbf{x}_0$ and $\mathbf{y}$ as input). Let us simply use $\boldsymbol{\Sigma}_{prior}$ and $\boldsymbol{\Sigma}_{post}$ hereafter to refer to $\boldsymbol{\Sigma}_{prior}(\mathbf{y}; \psi)$ and $\boldsymbol{\Sigma}_{post}(\mathbf{x}_0, \mathbf{y}; \phi)$ for concise notation. Then, with the direct sampling property in the forward path $\mathbf{x}_t = \sqrt{\bar{\alpha}_t}\mathbf{x}_0 + \sqrt{1 - \bar{\alpha}_t}\boldsymbol{\epsilon}$ at arbitrary timestep $t$ where $\boldsymbol{\epsilon} \sim q_\phi(\boldsymbol{\epsilon}|\mathbf{x}_0, \mathbf{y})$, and assuming the*

*reverse process has the same covariance as the true forward process posterior conditioned on $\mathbf{x}_0$, by utilizing the conditional DDPM $\boldsymbol{\epsilon}_\theta(\mathbf{x}_t, \mathbf{y}, t)$ as the noise estimator of the true noise $\boldsymbol{\epsilon}$, we have the modified ELBO, $-\mathcal{L}(\theta, \phi, \psi)$, associated with (9):*

$$\mathcal{L}(\theta, \phi, \psi) = \underbrace{\frac{\bar{\alpha}_T}{2}\mathbb{E}_{\mathbf{x}_0}\|\mathbf{x}_0\|^2_{\boldsymbol{\Sigma}^{-1}_{post}} + \frac{1}{2}\log|\boldsymbol{\Sigma}_{post}|}_{\text{Latent Regularization (LR) terms}} + \underbrace{\sum_{t=1}^{T}\gamma_t\mathbb{E}_{(\mathbf{x}_0,\mathbf{y}),\boldsymbol{\epsilon}\sim\mathcal{N}(\mathbf{0},\boldsymbol{\Sigma}_{post})}\|\boldsymbol{\epsilon} - \boldsymbol{\epsilon}_\theta(\mathbf{x}_t, \mathbf{y}, t)\|^2_{\boldsymbol{\Sigma}^{-1}_{post}}}_{\text{Denoising Matching (DM) terms}}$$

$$+ \underbrace{\frac{1}{2}\Big(\log\frac{|\boldsymbol{\Sigma}_{prior}|}{|\boldsymbol{\Sigma}_{post}|} + tr(\boldsymbol{\Sigma}^{-1}_{prior}\boldsymbol{\Sigma}_{post})\Big)}_{\text{Prior Matching (PM) terms}} + C,$$

*where $\gamma_t = \begin{cases} \frac{\beta_t^2}{2\sigma_t^2\alpha_t(1-\bar{\alpha}_t)}, & t > 1 \\ \frac{1}{2\alpha_1}, & t = 1 \end{cases}$ are weighting factors, $\|\mathbf{x}\|^2_{\boldsymbol{\Sigma}^{-1}} = \mathbf{x}^T\boldsymbol{\Sigma}^{-1}\mathbf{x}$, $\sigma_t^2 = \frac{1-\bar{\alpha}_{t-1}}{1-\bar{\alpha}_t}\beta_t$ and $C$ is some constant not depending on learnable parameters $\theta, \phi, \psi$. some constant not depending on the learnable parameters $\theta, \phi,$ and $\psi$.*

*Derivation:*

Recall our proposed lower bound in (9) to incorporate the conditional DDPM into the VAE framework is given as:

$$\mathbb{E}_{q_\phi(\boldsymbol{\epsilon}|\mathbf{x}_0,\mathbf{y})}\Bigg[\underbrace{\mathbb{E}_{q(\mathbf{x}_{1:T}|\mathbf{x}_0)}\bigg[\frac{p_\theta(\mathbf{x}_{0:T}|\mathbf{y})}{q(\mathbf{x}_{1:T}|\mathbf{x}_0)}\bigg]}_{-\mathcal{L}(\theta,\phi)}\Bigg] - D_{\text{KL}}\big(q_\phi(\boldsymbol{\epsilon}|\mathbf{x}_0,\mathbf{y})||p_\psi(\boldsymbol{\epsilon}|\mathbf{y})\big). \tag{16}$$

Note that as assumed in standard DDPMs, the forward diffusion process gradually corrupts the data distribution into the prior distribution, which can be achieved by carefully designing the variance schedule for the forward pass, i.e., $\{\beta_t\}_{t=1}^T$, such that $\mathbf{x}_T \to \boldsymbol{\epsilon}$ (as a result of $\bar{\alpha}_T \to 0$). More specifically, the $q(\mathbf{x}_T|\mathbf{x}_0)$ of the forward diffusion process converges in distribution to the approximate posterior $q_\phi(\boldsymbol{\epsilon}|\mathbf{x}_0, \mathbf{y})$ from the posterior encoder $\phi$. Then, the term $\mathcal{L}(\theta, \phi)$ in (16) suggests training a conditional diffusion model $\theta$ to reverse the diffusion trajectory from the estimated distribution of $\boldsymbol{\epsilon}$ given by the posterior encoder $\phi$ back to the target data distribution of $\mathbf{x}_0$.

The $-$ELBO $\mathcal{L}(\theta, \phi)$ can be shown to be expanded as (Luo, 2022; Sohl-Dickstein et al., 2015; Ho et al., 2020):

$$\begin{aligned} \mathcal{L}(\theta, \phi) &:= -\mathbb{E}_{q(\mathbf{x}_{1:T}|\mathbf{x}_0)}\bigg[\log\frac{p_\theta(\mathbf{x}_{0:T}|\mathbf{y})}{q(\mathbf{x}_{1:T}|\mathbf{x}_0)}\bigg] \\ &= -\mathbb{E}_{q(\mathbf{x}_{1:T}|\mathbf{x}_0)}\bigg[\log\frac{p(\mathbf{x}_T)\prod_{t=1}^T p_\theta(\mathbf{x}_{t-1}|\mathbf{x}_t,\mathbf{y})}{\prod_{t=1}^T q(\mathbf{x}_t|\mathbf{x}_{t-1})}\bigg] \\ &= -\mathbb{E}_{q(\mathbf{x}_{1:T}|\mathbf{x}_0)}\bigg[\log\frac{p(\mathbf{x}_T)p_\theta(\mathbf{x}_0|\mathbf{x}_1,\mathbf{y})\prod_{t=2}^T p_\theta(\mathbf{x}_{t-1}|\mathbf{x}_t,\mathbf{y})}{q(\mathbf{x}_1|\mathbf{x}_0)\prod_{t=2}^T q(\mathbf{x}_t|\mathbf{x}_{t-1})}\bigg] \\ &= -\mathbb{E}_{q(\mathbf{x}_{1:T}|\mathbf{x}_0)}\big[\log p_\theta(\mathbf{x}_0|\mathbf{x}_1,\mathbf{y})\big] - \mathbb{E}_{q(\mathbf{x}_{1:T}|\mathbf{x}_0)}\bigg[\log\frac{p(\mathbf{x}_T)}{q(\mathbf{x}_T|\mathbf{x}_0)}\bigg] - \sum_{t=2}^T\mathbb{E}_{q(\mathbf{x}_{1:T}|\mathbf{x}_0)}\bigg[\log\frac{p_\theta(\mathbf{x}_{t-1}|\mathbf{x}_t,\mathbf{y})}{q(\mathbf{x}_{t-1}|\mathbf{x}_t,\mathbf{x}_0)}\bigg] \\ &= -\mathbb{E}_{q(\mathbf{x}_1|\mathbf{x}_0)}\big[\log p_\theta(\mathbf{x}_0|\mathbf{x}_1,\mathbf{y})\big] - \mathbb{E}_{q(\mathbf{x}_T|\mathbf{x}_0)}\bigg[\log\frac{p(\mathbf{x}_T)}{q(\mathbf{x}_T|\mathbf{x}_0)}\bigg] - \sum_{t=2}^T\mathbb{E}_{q(\mathbf{x}_t,\mathbf{x}_{t-1}|\mathbf{x}_0)}\bigg[\log\frac{p_\theta(\mathbf{x}_{t-1}|\mathbf{x}_t,\mathbf{y})}{q(\mathbf{x}_{t-1}|\mathbf{x}_t,\mathbf{x}_0)}\bigg] \\ &= \mathcal{L}_0 + \mathcal{L}_T + \sum_{t=2}^T\mathcal{L}_{t-1}, \end{aligned} \tag{17}$$

where

$$\mathcal{L}_0 := -\mathbb{E}_{q(\mathbf{x}_1|\mathbf{x}_0)}\big[\log p_\theta(\mathbf{x}_0|\mathbf{x}_1,\mathbf{y})\big], \tag{18}$$

$$\mathcal{L}_{t-1} := \mathbb{E}_{q(\mathbf{x}_t|\mathbf{x}_0)}\big[\mathcal{D}_{\text{KL}}\big(q(\mathbf{x}_{t-1}|\mathbf{x}_t,\mathbf{x}_0)||p_\theta(\mathbf{x}_{t-1}|\mathbf{x}_t,\mathbf{y})\big)\big], \tag{19}$$

$$\mathcal{L}_T := \mathcal{D}_{\text{KL}}\big(q(\mathbf{x}_T|\mathbf{x}_0)||p(\mathbf{x}_T)\big). \tag{20}$$

According to Lee et al. (2022), the terms of the loss function for training the noise estimator network $\theta$ of the conditional DDPM for an arbitrary $\epsilon \sim \mathcal{N}(\mathbf{0}, \boldsymbol{\Sigma})$ can be explicitly written as:

$$\mathcal{L}_0 = \frac{1}{2} \log\left((2\pi\beta_1)^d|\boldsymbol{\Sigma}|\right) + \frac{1}{2\alpha_1}\mathbb{E}_{\mathbf{x}_0, \epsilon \sim \mathcal{N}(\mathbf{0}, \boldsymbol{\Sigma})}\|\epsilon - \epsilon_\theta(\mathbf{x}_1, \mathbf{y}, 1)\|_{\boldsymbol{\Sigma}^{-1}}^2,$$

$$\begin{aligned}\mathcal{L}_{t-1} &= \frac{\beta_t}{2\alpha_t(1 - \bar{\alpha}_{t-1})}\mathbb{E}_{\mathbf{x}_0, \epsilon \sim \mathcal{N}(\mathbf{0}, \boldsymbol{\Sigma})}\|\epsilon - \epsilon_\theta(\mathbf{x}_t, \mathbf{y}, t)\|_{\boldsymbol{\Sigma}^{-1}}^2 \\ &= \frac{\beta_t^2}{2\sigma_t^2\alpha_t(1 - \bar{\alpha}_t)}\mathbb{E}_{\mathbf{x}_0, \epsilon \sim \mathcal{N}(\mathbf{0}, \boldsymbol{\Sigma})}\|\epsilon - \epsilon_\theta(\mathbf{x}_t, \mathbf{y}, t)\|_{\boldsymbol{\Sigma}^{-1}}^2,\end{aligned} \tag{21}$$

$$\mathcal{L}_T = \frac{\bar{\alpha}_T}{2}\mathbb{E}_{\mathbf{x}_0}\|\mathbf{x}_0\|_{\boldsymbol{\Sigma}^{-1}}^2 - \frac{d}{2}(\bar{\alpha}_T + \log(1 - \bar{\alpha}_T)),$$

with $\bar{\alpha}_t := \prod_{i=1}^t \alpha_i$ and $\alpha_t := 1 - \beta_t$ for $t = 1, \ldots, T$ where $\{\beta_t\}_{t=1}^T$ is the noise variance schedule as a hyperparameter, $d$ is the parameter freedom and $\sigma_t^2 = \frac{1 - \bar{\alpha}_{t-1}}{1 - \bar{\alpha}_t}\beta_t$.

In our case, we have assumed modeling of the posterior distribution where the $\epsilon$ is sampled from as the zero-mean Gaussian $\epsilon \sim \mathcal{N}(\mathbf{0}, \boldsymbol{\Sigma}_{\text{post}})$ where the covariance $\boldsymbol{\Sigma}_{\text{post}} := \boldsymbol{\Sigma}_{\text{post}}(\mathbf{x}_0, \mathbf{y}; \phi)$ is estimated by the Posterior Net $\phi$, taking both the ground truth data $\mathbf{x}_0$ and the conditioner $\mathbf{y}$ as input. By directly plugging in $\boldsymbol{\Sigma} = \boldsymbol{\Sigma}_{\text{post}}$ for each term in (21), we obtain:

$$\mathcal{L}(\theta, \phi) = \frac{\bar{\alpha}_T}{2}\mathbb{E}_{\mathbf{x}_0}\|\mathbf{x}_0\|_{\boldsymbol{\Sigma}_{\text{post}}^{-1}}^2 + \frac{1}{2}\log|\boldsymbol{\Sigma}_{\text{post}}| + \sum_{t=1}^T \gamma_t \mathbb{E}_{(\mathbf{x}_0, \mathbf{y}), \epsilon \sim \mathcal{N}(\mathbf{0}, \boldsymbol{\Sigma}_{\text{post}})}\|\epsilon - \epsilon_\theta(\underbrace{\sqrt{\bar{\alpha}_t}\mathbf{x}_0 + \sqrt{1 - \bar{\alpha}_t}\epsilon}_{\mathbf{x}_t}, \mathbf{y}, t)\|_{\boldsymbol{\Sigma}_{\text{post}}^{-1}}^2 + C, \tag{22}$$

where

$$\gamma_t = \begin{cases} \frac{\beta_t^2}{2\sigma_t^2\alpha_t(1 - \bar{\alpha}_t)}, & t > 1 \\ \frac{1}{2\alpha_1}, & t = 1 \end{cases}$$

and $C$ is some constant not depending on the learnable parameters.

For the prior matching term in (16), we can utilize the analytic form of the KL divergence between two Gaussians which leads to:

$$D_{\text{KL}}\left(q_\phi(\epsilon|\mathbf{x}_0, \mathbf{y})||p_\psi(\epsilon|\mathbf{y})\right) = \frac{1}{2}\left(\log\frac{|\boldsymbol{\Sigma}_{\text{prior}}|}{|\boldsymbol{\Sigma}_{\text{post}}|} + \text{tr}(\boldsymbol{\Sigma}_{\text{prior}}^{-1}\boldsymbol{\Sigma}_{\text{post}})\right), \tag{23}$$

where the covariances $\boldsymbol{\Sigma}_{\text{prior}} := \boldsymbol{\Sigma}_{\text{prior}}(\mathbf{y}; \psi)$ and $\boldsymbol{\Sigma}_{\text{post}} := \boldsymbol{\Sigma}_{\text{post}}(\mathbf{x}_0, \mathbf{y}; \phi)$.

Combining (22) and (23), we have obtained the $-$ELBO $\mathcal{L}(\theta, \phi, \psi)$ of Proposition 3.2.

## C. Implementation Details

### C.1. Algorithms

---

**Algorithm 1** Training of RestoreGrad

**for** $i = 0, 1, 2..., N_{iter}$ **do**
    Sample $(\mathbf{x}_0, \mathbf{y}) \sim q_{\text{data}}(\mathbf{x}_0, \mathbf{y})$
    $\boldsymbol{\Sigma}_{\text{prior}} \leftarrow$ Prior Net$(\mathbf{y}; \psi)$
    $\boldsymbol{\Sigma}_{\text{post}} \leftarrow$ Posterior Net$(\mathbf{x}_0, \mathbf{y}; \phi)$
    Sample $\epsilon \sim \mathcal{N}(\mathbf{0}, \boldsymbol{\Sigma}_{\text{post}})$ and $t \sim \mathcal{U}(\{1, \ldots, T\})$
    $\mathbf{x}_t = \sqrt{\bar{\alpha}_t}\mathbf{x}_0 + \sqrt{1 - \bar{\alpha}_t}\epsilon$
    $\mathcal{L}_{\text{LR}} = \bar{\alpha}_T||\mathbf{x}_0||_{\boldsymbol{\Sigma}_{\text{post}}^{-1}}^2 + \log|\boldsymbol{\Sigma}_{\text{post}}|$
    $\mathcal{L}_{\text{DM}} = ||\epsilon - \epsilon_\theta(\mathbf{x}_t, \mathbf{y}, t)||_{\boldsymbol{\Sigma}_{\text{post}}^{-1}}^2$
    $\mathcal{L}_{\text{PM}} = \log\frac{|\boldsymbol{\Sigma}_{\text{prior}}|}{|\boldsymbol{\Sigma}_{\text{post}}|} + \text{tr}(\boldsymbol{\Sigma}_{\text{prior}}^{-1}\boldsymbol{\Sigma}_{\text{post}})$
    Update $\theta, \psi, \phi$ with $\nabla_{\theta, \psi, \phi}\, \eta\mathcal{L}_{\text{LR}} + \mathcal{L}_{\text{DM}} + \lambda\mathcal{L}_{\text{PM}}$
**end for**

---

**Algorithm 2** Sampling of RestoreGrad

$\boldsymbol{\Sigma}_{\text{prior}} \leftarrow$ Prior Net$(\mathbf{y}; \psi)$
Sample $\mathbf{x}_T \sim \mathcal{N}(\mathbf{0}, \boldsymbol{\Sigma}_{\text{prior}})$
**for** $t = T, T - 1, ..., 1$ **do**
    **if** $t > 0$ **then**
        Sample $\epsilon \sim \mathcal{N}(\mathbf{0}, \boldsymbol{\Sigma}_{\text{prior}})$
    **else**
        $\epsilon = 0$
    **end if**
    $\mathbf{x}_{t-1} = \frac{1}{\sqrt{\alpha_t}}\left(\mathbf{x}_t - \frac{1 - \alpha_t}{\sqrt{1 - \bar{\alpha}_t}}\epsilon_\theta(\mathbf{x}_t, \mathbf{y}, t)\right) + \sigma_t\epsilon$
**end for**
**return** $\mathbf{x}_0$

---

## C.2. Experiments on Speech Enhancement (SE)

### C.2.1. DATASET

We used the VoiceBank+DEMAND dataset (Valentini-Botinhao et al., 2016) with the same experimental setup as in previous work (Pascual et al., 2017; Phan et al., 2020; Strauss & Edler, 2021; Lu et al., 2022) to perform a direct comparison. The clean speech and noise recordings were provided from the VoiceBank corpus (Veaux et al., 2013) and the Diverse Environments Multichannel Acoustic Noise Database (DEMAND) (Thiemann et al., 2013), respectively, each recorded with sampling rate of 48kHz. Noisy speech inputs used for training were composed by mixing the two datasets with four signal-to-noise ratio (SNR) settings from {0, 5, 10, 15} dB, using 10 types of noise (2 artificially generated + 8 real recorded from the DEMAND dataset) and 28 speakers from the Voice Bank corpus. The test set inputs were made with four SNR settings different from the training set, i.e., {2.5, 7.5, 12.5, 17.5} dB, using the remaining 5 noise types from DEMAND and 2 speakers from the VoiceBank corpus. There are totally 11,527 utterances for training and 824 for testing. Note that the speaker and noise classes were uniquely selected for the training and test sets. The dataset is publicly available at: `https://datashare.ed.ac.uk/handle/10283/2826`. In our experiments, the audio steams were resampled to 16kHz sampling rate.

### C.2.2. MODEL ARCHITECTURE

**Baseline DDPM-based SE Model:** The baseline SE model considered in this work, i.e., CDiffuSE (Lu et al., 2022), performs enhancement in the time domain. We utilized the CDiffuSE base model, which has approximately 4.28M learnable parameters, from the implementation at: `https://github.com/neillu23/CDiffuSE`. The model is implemented based on DiffWave (Kong et al., 2021), a versatile diffusion probabilistic model for conditional and unconditional waveform generation. The basic model structure of CDiffuSE is similar to that of DiffWave. However, since the target task is SE, CDiffuSE uses the noisy spectral features as the conditioner, rather than the clean Mel-spectral features used in DiffWave utilized for vocoders. After the reverse process is completed, the enhanced waveform further combine the observed noisy signal $\mathbf{y}$ with the ratio 0.2 to recover the high frequency speech in the final enhanced waveform, as suggested in Abd El-Fattah et al. (2008); Defossez et al. (2020).

**PriorGrad:** We implemented the PriorGrad (Lee et al., 2022) on top of the CDiffuSE model by using a data-dependent prior $\mathcal{N}(\mathbf{0}, \boldsymbol{\Sigma}_{\mathrm{y}})$, where $\boldsymbol{\Sigma}_{\mathrm{y}}$ is the covariance of the prior distribution computed based on using the mel-spectrogram of the noisy input $\mathbf{y}$. Following the application to vocoder in Lee et al. (2022), we leveraged a normalized frame-level energy of the mel-spectrogram for acquiring data-dependent prior, exploiting the fact that the spectral energy contains an exact correlation to the waveform variance (by Parseval's theorem (Stoica et al., 2005)). More specifically, we computed the frame-level energy by taking the square root of the sum of $\exp(\mathbf{Y})$ over the frequency axis for each time frame, where $\mathbf{Y}$ is the mel-spectrogram of the noisy input $\mathbf{y}$ from the training data. We then normalized the frame-level energy to a range of $(0, 1]$ to acquire the data-dependent diagonal variance $\boldsymbol{\Sigma}_Y$. Then we upsampled $\boldsymbol{\Sigma}_Y$ in the frame level to $\boldsymbol{\Sigma}_y$ in the waveform-level using the given hop length of computing the mel-spectrogram. We imposed the minimum standard deviation of the prior to 0.1 through clipping to ensure numerical stability during training, as suggested in Lee et al. (2022).

**Prior Net and Posterior Net for RestoreGrad:** The additional encoder modules for the RestoreGrad adopt the ResNet-20 architect (He et al., 2016) using the implementation from: `https://github.com/akamaster/pytorch_resnet_cifar10`. We suitably modified the original 2-D convolutions in ResNet-20 to 1-D convolutions for waveform processing. The modified ResNet-20 model has only 93K learnable parameters (only 2% of the size of CDiffuSE model). The Prior Net takes the noisy speech waveform $\mathbf{y}$ as input, while the Posterior Net takes both the clean and noisy waveforms, $\mathbf{x}_0$ and $\mathbf{y}$, as input, which are concatenated along the channel dimension. We employed the exponential nonlinearity at the network output for estimating the variances of the prior and posterior distributions.

### C.2.3. OPTIMIZATION AND INFERENCE

We used the same configurations of CDiffuSE (Base) (Lu et al., 2022) for optimizing all the models, where the batch size was 16, the Adam optimizer was used with a learning rate of $2 \times 10^{-4}$, and the diffusion steps $T = 50$ with linearly spaced $\beta_t \in [10^{-4}, 0.035]$. For RestoreGrad, we imposed the minimum standard deviation $\sigma_{\min} = 0.1$ by adding it to the output of the Prior Net and Posterior Net to ensure stability during training. The fast sampling scheme in Kong et al. (2021) was used in the reverse processes with 6-step inference schedule $\beta_t^{\mathrm{infer}} = [10^{-4}, 10^{-3}, 0.01, 0.05, 0.2, 0.35]$. The models were trained on one NVIDIA Tesla V100 GPU of 32 GB CUDA memory and finished training for 96 epochs in 1 day.

### C.2.4. EVALUATION METRICS

**PESQ:** a speech quality measure using the wide-band version recommended in ITU-T P.862.2 (ITU-T Rec. P.862.2, 2005). It basically models the mean opinion scores (MOS) that cover a scale from 1 (bad) to 5 (excellent). We used the Python-based PESQ implementation from: `https://github.com/ludlows/python-pesq`.

**SI-SNR:** a variant of the conventional SNR measure taking into account the scale-invariance of audio signals. The SI-SDR is a more robust and meaningful metric than the traditional SNR for measuring speech quality. A higher SI-SNR score indicates better perceptual speech quality. We adopted the SI-SNR implementation from: `https://lightning.ai/docs/torchmetrics/stable/audio/scale_invariant_signal_noise_ratio.html`.

**SSNR:** an SNR measure, instead of working on the whole signal, that calculates the average of the SNR values of short segments (segment length = 30 msec, 75% overlap, $SNR_{min} = -10$ dB, $SNR_{max} = 35$ dB). We use the Python-based SSNR implementation from: `https://github.com/schmiph2/pysepm`.

**CSIG:** The mean opinion score (MOS) prediction of the signal distortion (from 1 to 5, the higher the better) (Hu & Loizou, 2007). We used the implementation from: `https://github.com/schmiph2/pysepm`.

**CBAK:** MOS prediction of the intrusiveness of background noises (from 1 to 5, the higher the better) (Hu & Loizou, 2007). We used the implementation from: `https://github.com/schmiph2/pysepm`.

**COVL:** MOS prediction of the overall effect (from 1 to 5, the higher the better) (Hu & Loizou, 2007). We used the implementation from: `https://github.com/schmiph2/pysepm`.

### C.3. Experiments on Image Restoration (IR)

#### C.3.1. DATASETS

We used three standard benchmark image restoration datasets considering adverse weather conditions of snow, heavy rain with haze, and raindrops on the camera sensor, following Özdenizci & Legenstein (2023).

**Snow100K (Liu et al., 2018):** a dataset for evaluation of image desnowing models. The images are split into approximately equal sizes of three Snow100K-S/M/L sub-test sets (with 16,611/16,588/16,801 samples, respectively), indicating the synthetic snow strength imposed via snowflake sizes (light/mid/heavy). We used the Snow100K-L sub-test set for evaluation. The dataset can be downloaded from: `https://sites.google.com/view/yunfuliu/desnownet`.

**Outdoor-Rain (Li et al., 2019):** a dataset of simultaneous rain and fog which exploits a physics-based generative model to simulate not only dense synthetic rain streaks, but also incorporating more realistic scene views, constructing an inverse problem of simultaneous image deraining and dehazing. We used the test set, denoted in Li et al. (2019) as Test1, which is of size 750 for quantitative evaluations. The dataset can be accessed at: `https://github.com/liruoteng/HeavyRainRemoval`.

**RainDrop (Qian et al., 2018):** a dataset of images with raindrops introducing artifacts on the camera sensor and obstructing the view. It consists of 861 training images with synthetic raindrops, and a test set of 58 images dedicated for quantitative evaluations, denoted in Qian et al. (2018) as RainDrop-A. The dataset is provided at: `https://github.com/rui1996/DeRaindrop`.

In addition, we also used the composite dataset for multi-weather IR model training:

**AllWeather (Valanarasu et al., 2022):** is a dataset of 18,069 samples composed of subsets of training images from the training sets of the three datasets above, in order to create a balanced training set across three weather conditions with a similar approach to Li et al. (2020). The dataset is publicly available at: `https://github.com/jeya-maria-jose/TransWeather`.

#### C.3.2. MODEL ARCHITECTURE

**Baseline DDPM-based IR Models:** The baseline IR models considered in this work, i.e., the RainDropDiff and WeatherDiff from Özdenizci & Legenstein (2023), perform patch-based diffusive restoration of the images. The models perform diffusion process at the patch level, where overlapping $p \times p$ patches are taken as input. When sampling, all $p \times p$ patches extracted from the image with a hop size $r$ are processed by the DDPM model, utilizing the mean estimated noise based sampling updates for the overlapping pixels to synthesize the clean image. In this work, we considered $p = 64$ and $r = 16$, which

correspond to the RainDropDiff$_{64}$ and WeatherDiff$_{64}$ models (with 110M and 82 M learnable parameters, respectively) provided by the authors at: `https://github.com/IGITUGraz/WeatherDiffusion`.

**Prior Net and Posterior Net for RestoreGrad:** The additional encoder modules for the RestoreGrad adopt the ResNet-20 architect (He et al., 2016) using the implementation from: `https://github.com/akamaster/pytorch_resnet_cifar10`. The ResNet-20 model has 0.27M learnable parameters, which is less than 0.3% of the size of RainDropDiff and WeatherDiff. The Prior Net takes the noisy image $\mathbf{y}$ as input, while the Posterior Net takes both the clean and noisy images, $\mathbf{x}_0$ and $\mathbf{y}$, as input, which are concatenated along the channel dimension. We employed the exponential nonlinearity at the network output for estimating the variances of the prior and posterior distributions.

### C.3.3. Optimization and Inference

We used the same configurations of Özdenizci & Legenstein (2023) for optimizing all the models, except the batch size was 4 instead of 16 due to GPU memory limitation. The Adam optimizer with a fixed learning rate of $2 \times 10^{-5}$ was used for training models without weight decay, and an exponential moving average with a weight of 0.999 was applied during parameter updates. The number of diffusion steps was $T = 1000$ and the noise schedule was $\beta_t \in [10^{-4}, 0.02]$, linearly spaced. For inference, we used the sampling scheme with 10 timesteps for each model that we trained on our own. We did not use the deterministic implicit sampling scheme as in Özdenizci & Legenstein (2023) for our RestoreGrad-based DDPM models as we found using the normal stochastic sampling scheme actually works better. The models were trained on 2 NVIDIA Tesla V100 GPU of 32 GB CUDA memory and finished training for 9,261 epochs on the RainDrop dataset in 12 days and 887 epochs on the AllWeather dataset in 21 days.

### C.3.4. Evaluation Metrics

**PSNR:** a non-linear full-reference metric that compares the pixel values of the original reference image to the values of the degraded image based on the mean squared error (Huynh-Thu & Ghanbari, 2008). A higher PSNR indicates better reconstruction quality of images in terms of distortion. PSNR can be calculated for the different color spaces. We followed Özdenizci & Legenstein (2023) to compute PSNR based on the luminance channel Y of the YCbCr color space. We used the implementation form `https://github.com/JingyunLiang/SwinIR` for PSNR calculation.

**SSIM:** a non-linear full-reference metric compares the luminance, contrast and structure of the original and degraded image (Wang et al., 2004). It provides a value from 0 to 1, the closer the score is to 1, the more similar the degraded image is to the reference image. We followed Özdenizci & Legenstein (2023) to compute SSIM based on the luminance channel Y of the YCbCr color space. We used the implementation form `https://github.com/JingyunLiang/SwinIR` for SSIM.

**LPIPS** (Zhang et al., 2018) and **FID** (Heusel et al., 2017): to provide better quantification of perceptual quality over the traditional distortion measures of PSNR and SSIM (Blau & Michaeli, 2018; Freirich et al., 2021). For the LPIPS we used the implementation from `https://github.com/richzhang/PerceptualSimilarity`, and for FID we used the implementation from `https://github.com/chaofengc/IQA-PyTorch`. In both metrics, a lower score indicates better perceptual quality of the restored image.

### C.4. Experiments on Generalization to OOD and Realistic Data

#### C.4.1. Datasets

The additional datasets considered for experiments on realistic data for the IR task are:

**RainDS-Real (Qian et al., 2018):** is the raindrop removal test subset of the RainDS dataset presented in Qian et al. (2018). It consists of 98 real-world captured raindrop obstructed images. The dataset is publicly available at: `https://github.com/Songforrr/RainDS_CCN`.

**Snow100K-Real (Liu et al., 2018):** is the subset of the Snow100K dataset (Liu et al., 2018) that consists of 1,329 realistic snowy images for testing real-world restoration cases. The dataset can be accessed at: `https://sites.google.com/view/yunfuliu/desnownet`.

The additional dataset considered for experiments on OOD data of the SE task is:

**CHiME-3 (Barker et al., 2017):** is a 6-channel microphone recording of talkers speaking in a noisy environment, sampled at 16 kHz. It consists of 7,138 and 1,320 simulated utterances for training and testing, respectively, which are generated

by artificially mixing clean speech data with noisy backgrounds of four types, i.e. cafe, bus, street, and pedestrian area. In this paper, we only take the 5-th channel recordings for the experiments. The dataset information can be found at: `https://www.chimechallenge.org/challenges/chime3/data`.

### C.4.2. EVALUATION METRICS

The additional evaluation metric used in the corresponding section is:

**NIQE:** is a reference-free quality assessment of real-world restoration performance introduced by Mittal et al. (2012) which measures the naturalness of a given image without using any reference image. A lower NIQE score indicates better perceptual image quality. We used the NIQE implementation from: `https://github.com/chaofengc/IQA-PyTorch`.

### C.5. Applications to Other IR tasks

#### C.5.1. DATASETS

The datasets considered for experiments on image deblurring and super-resolution tasks are:

**RealBlur (Rim et al., 2020):** a large-scale dataset of real-world blurred images and ground truth sharp images for learning and benchmarking single image deblurring methods. The images were captured both in the camera raw and JPEG formats, leading to two datasets: *RealBlur-R* from the raw images and *RealBlur-J* from the JPEG images. Each training set consists of 3,758 image pairs and each test set consists of 980 image pairs. The dataset can be downloaded from: `https://cg.postech.ac.kr/research/realblur/`.

**DIV2K (Agustsson & Timofte, 2017; Timofte et al., 2017):** a dataset of 2K resolution high quality images collected from the Internet as part of the NTIRE 2017 super-resolution challenge. There are 800, 100, and 100 images for training, validation, and testing, respectively. The dataset provides $\times 2$, $\times 3$, and $\times 4$ downscaled images with bicubic and unknown downgrading operations. The dataset can be downloaded from: `https://data.vision.ee.ethz.ch/cvl/DIV2K/`.

#### C.5.2. MODEL ARCHITECTURE

The baseline conditional DDPM (cDDPM) implements the same architecture as the patch-based denoising diffusion model of WeatherDiff (Özdenizci & Legenstein, 2023). The Prior Net and Posterior Net of RestoreGrad also adopt the same ResNet models as in the IR experiments under adverse weather conditions. For more details please refer to Appendix C.3.2.

#### C.5.3. OPTIMIZATION AND INFERENCE

The models were optimized and inferenced using the same configurations and settings as given in Appendix C.3.3 for the IR experiments under adverse weather conditions. The models were trained on 2 NVIDIA Tesla V100 GPU of 32 GB CUDA memory and finished training for 853 epochs on the RealBlur-{R,J} dataset each in 5 days and 2000 epochs on the DIV2K-{$\times 2$,$\times 4$} dataset each in 3 days.

## D. Additional Experimental Results

### D.1. Additional Results on SE

**Model Learning Performance in Terms of Other Metrics:** In addition to the results evaluated by PESQ and COVL in Figure 2, we provide the learning curves in terms of the CSIG, CBAK, and SI-SNR metrics in Figure 8, to further support the advantages of RestoreGrad over the baseline DDPM and PriorGrad for improved training behavior and efficiency.

**Performance with Using Different Numbers of Inference Steps:** In Figure 9, we show how the trained diffusion models perform with respect to using different numbers of reverse steps for inference. Specifically, in each case of CDiffuSE, PriorGrad, and RestoreGrad, we trained the model for 96 epochs and then inferenced with $S \in \{3,4,5\}$ reverse steps to compare with the originally adopted $S = 6$ steps in Lu et al. (2022). We used $\beta_t^{\text{infer}} = [10^{-4}, 10^{-3}, 0.05, 0.2, 0.35]$ for $S = 5$, $\beta_t^{\text{infer}} = [10^{-4}, 0.05, 0.2, 0.35]$ for $S = 4$, and $\beta_t^{\text{infer}} = [0.05, 0.2, 0.35]$ for $S = 3$. These choices were selected from the subsets of the original noise schedule for $S = 6$, i.e., $\beta_t^{\text{infer}} = [10^{-4}, 10^{-3}, 0.01, 0.05, 0.2, 0.35]$, that resulted in best performance of the models. For the figure we can see that as $S$ becomes smaller, the baseline CDiffuSE degrades

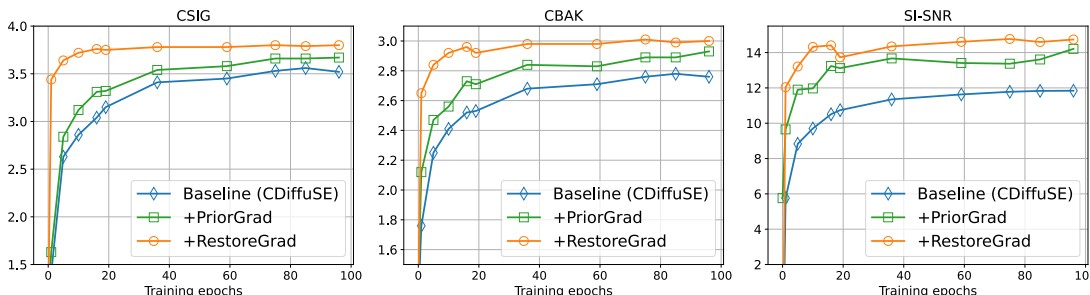

Figure 8. Model learning performance in terms of CSIG, CBAK, and SI-SNR metrics. Improved training behavior of RestoreGrad over CDiffSE and PriorGrad is observed among all metrics.

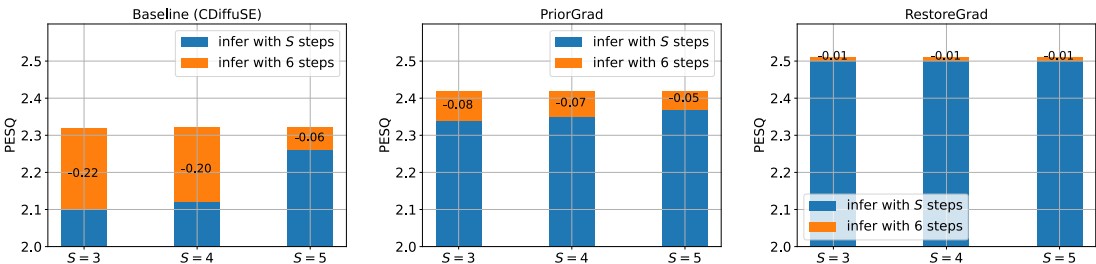

Figure 9. Effect of using reduced numbers of sampling steps in inference on the SE performance, in terms of PESQ. RestoreGrad demonstrates the strongest resistance to the reduction in reverse sampling steps for inference.

Table 7. Performance comparison of RestoreGrad with the baseline DDPM (CDiffSE) and PriorGrad for using various numbers of sampling steps $S$ during inference.

| Methods | SI-SNR ↑ | | | | CSIG ↑ | | | | CBAK ↑ | | | | COVL ↑ | | | |
|---|---|---|---|---|---|---|---|---|---|---|---|---|---|---|---|---|
| | $S$=6 | $S$=5 | $S$=4 | $S$=3 | $S$=6 | $S$=5 | $S$=4 | $S$=3 | $S$=6 | $S$=5 | $S$=4 | $S$=3 | $S$=6 | $S$=5 | $S$=4 | $S$=3 |
| CDiffSE (Lu et al., 2022) | 11.84 | 11.46 | 11.32 | 11.28 | 3.52 | 3.44 | 3.15 | 3.13 | 2.76 | 2.72 | 2.64 | 2.63 | 2.89 | 2.82 | 2.60 | 2.58 |
| + PriorGrad (Lee et al., 2022) | 14.21 | 13.98 | 13.93 | 13.93 | 3.67 | 3.61 | 3.56 | 3.54 | 2.93 | 2.90 | 2.88 | 2.88 | 3.02 | 2.97 | 2.93 | 2.92 |
| + RestoreGrad (ours) | **14.74** | **14.66** | **14.64** | **14.65** | **3.80** | **3.77** | **3.75** | **3.75** | **3.00** | **2.99** | **2.99** | **2.99** | **3.14** | **3.12** | **3.11** | **3.11** |

*Best values are indicated with bold text.

considerably, while PriorGrad shows certain resistance, and RestoreGrad manages to maintain the high performance. We present more comparison in Table 7 in terms of SI-SNR, CSIG, CBAK, and COVL metrics. The results further support that RestoreGrad is much more robust to the reduction in sampling steps, achieving the best quality scores in all the metrics over the baseline DDPM and PriorGrad across all sampling steps considered.

**Visualizing the Learned Prior.** It would be interesting to see how the latent noise prior that has been learned by RestoreGrad looks like and how it compares with that of the PriorGrad. In Figure 10 we present an example of a randomly chosen noisy speech waveform and the corresponding latent noise $\boldsymbol{\Sigma}_y = \text{diag}\{\boldsymbol{\sigma}_y^2\}$ of PriorGrad and that of RestoreGrad (with $(\eta, \lambda) = (0.1, 0.5)$ for (11)). It can be seen that the variances of the pre-defined (PriorGrad) and learned (RestoreGrad) latent noise distributions are actually quite different, though both show the trend of following the variation of the conditioner signal level. This trend indicates that both latent distributions aim to better approximate the true signal distribution in a more informative manner for improved efficiency, as against the standard Gaussian prior used in the original DDPM. Note that in the RestoreGrad training, we have chosen a proper KL weight $\lambda$ so that the Prior Net distribution matches the Posterior Net distribution reasonably well without harming the reconstruction performance of the DDPM model. On the other hand, using a too large $\lambda$ might lead to a collapsed latent space as the optimization could have put too much emphasis on matching the prior and posterior distribution, discarding the contribution of the reconstruction loss term. In contrast, using a too small $\lambda$ might result in large discrepancy between the learned prior and posterior distributions, as also illustrated in Figure 10. Empirically, we found a naive choice of 1 works reasonably well and also for similar values, e.g., 0.5, 10, etc., as similarly observed in the VAE-type model of Kohl et al. (2018).

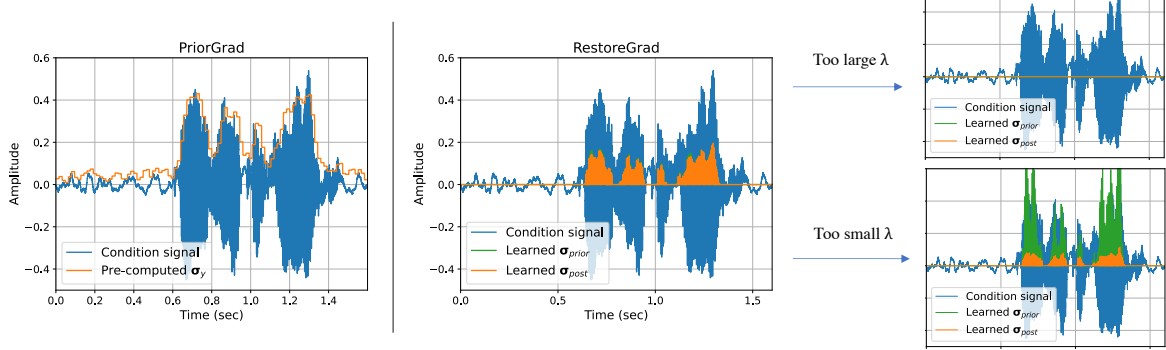

*Figure 10.* An example of learned latent distribution variances, $\mathbf{\Sigma}_{\text{prior}} = \text{diag}\{\boldsymbol{\sigma}^2_{\text{prior}}\}$ and $\mathbf{\Sigma}_{\text{post}} = \text{diag}\{\boldsymbol{\sigma}^2_{\text{post}}\}$ by RestoreGrad, and the effect of the KL weight $\lambda$ of the prior matching loss $\mathcal{L}_{\text{PM}}$ on the resulting latent distribution variances. The pre-computed variance of the handcrafted prior using PriorGrad is also presented for reference purposes.

**Comparison to Existing Waveform-Domain Generative SE Models:** In Table 8 we benchmark RestoreGrad with several generative SE approaches. Note that although RestoreGrad performs slightly inferior to DOSE, a recent SE model also based on DiffWave (Kong et al., 2021), it was actually achieved with $4.6\times$ fewer training epochs.

*Table 8.* Comparison with existing time-domain, generative SE models.

| Methods | PESQ↑ | CSIG↑ | CBAK↑ | COVL↑ |
|---|---|---|---|---|
| Unprocessed | 1.97 | 3.35 | 2.44 | 2.63 |
| SEGAN (Pascual et al., 2017) | 2.16 | 3.48 | 2.94 | 2.80 |
| DSEGAN (Phan et al., 2020) | 2.39 | 3.46 | 3.11 | 2.90 |
| SE-Flow (Strauss & Edler, 2021) | 2.28 | 3.70 | 3.03 | 2.97 |
| DOSE (Tai et al., 2023a) | **2.56** | **3.83** | **3.27** | **3.19** |
| CDiffuSE (Lu et al., 2022) | 2.44 | 3.66 | 2.83 | 3.03 |
| + RestoreGrad (ours) | 2.51 | 3.80 | 3.00 | 3.14 |

*Best values in bold and second best values underlined.

**Evaluation Using Automatic Speech Recognition (ASR):** Following Benita et al. (2024) who perform evaluation of diffusion-based speech generation using ASR, we evaluate the SE model as a front-end denoiser for ASR under noisy environments. To this end, we pre-process the noisy VoiceBand+DEMAND test data samples through the well-trained SE model and feed the denoised audio separately to two pre-trained ASR engines taken from the NVIDIA NeMo toolkit[1]: *Conformer-transducer-large* (Gulati et al., 2020) and *Citrinet-1024* (Majumdar et al., 2021). We report the word error rate (WER) and character error rate (CER) for each ASR engine outcome, where the lower WER / CER indicate better SE performance. The results are presented in Table 9 with all the SE models trained after 96 epochs, inferred using 6 steps. It is interesting to see that CDiffuSE and PriorGrad actually lead to worse performance than the unprocessed speech case for Citrinet ASR. Our RestoreGrad is able to achieve the lowest WER and CER for both ASR models, demonstrating its efficacy for enhancing machine learning capabilities under noisy environments.

**Enhanced Speech Examples:** We present several audio examples in Figure 11 to facilitate the comparison of the baseline DDPM and our RestoreGrad. It can be seen the RestoreGrad is able to recover a better speech signal closer to the target clean speech, which is also reflected by the higher PESQ scores obtained.

### D.2. Additional Results on IR

**Comparison to Existing IR Models on RainDrop Dataset:** We compare our method with existing IR models including AttentiveGAN (Qian et al., 2018), DuRN (Liu et al., 2019), RaindropAttn (Quan et al., 2019), IDT (Xiao et al., 2022) and RDDM (Liu et al., 2024) in Table 10 on the RainDrop dataset (Qian et al., 2018), where the models were all trained and

---

[1]https://github.com/NVIDIA/NeMo

*Table 9.* Following Benita et al. (2024) who perform evaluation of diffusion-based speech generation using ASR, we evaluate SE models on two ASR engines (Conformer, Citrinet) for the VoiceBand+DEMAND test set. The results further confirm the superiority of RestoreGrad over the baseline DDPM (CDiffuSE) and PriorGrad.

| | ASR: WER ↓ (%) / CER ↓ (%) | |
| --- | --- | --- |
| SE model | Conformer (Gulati et al., 2020) | Citrinet (Majumdar et al., 2021) |
| Unprocessed | 6.62 / 6.15 | 8.69 / 6.86 |
| CDiffuSE (Lu et al., 2022) | 6.55 / 6.01 | 9.77 / 7.41 |
| + PriorGrad (Lee et al., 2022) | 6.13 / 5.70 | 9.15 / 7.00 |
| + RestoreGrad (Ours) | **5.07 / 5.27** | **8.15 / 6.51** |

*Best values are indicated with bold text.

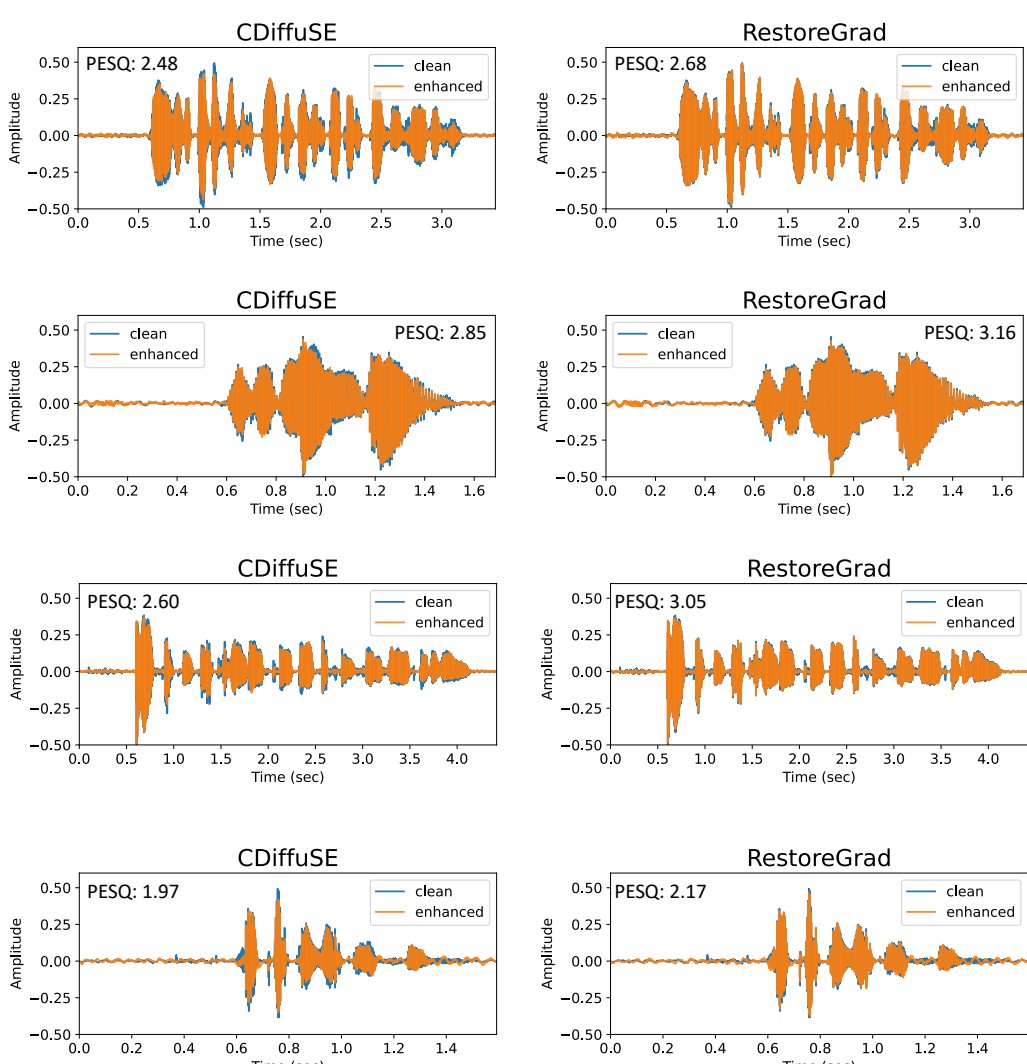

*Figure 11.* Enhanced speech examples of the baseline DDPM (CDiffuSE) and the proposed RestoreGrad for several noisy samples taken from the VoiceBank+DEMAND test set.

tested on the same training and test samples. The results of the compared models were taken from Özdenizci & Legenstein (2023) and Liu et al. (2024), where the baseline RainDropDiff was trained for 37,042 epochs. *Our RestoreGrad was only trained for 9,261 epochs (4× fewer than RainDropDiff), and has achieved higher PSNR and SSIM scores than RainDropDiff*

(here we report mean $\pm$ std of RestoreGrad based on results of 10 independent samplings). Moreover, our performance is comparable to the recent state-of-the-art approach of RDDM, further suggesting the potential of our method to effectively improve baseline DDPM approaches (e.g., RainDropDiff).

*Table 10.* Weather-specific (RainDrop dataset) model comparison.

| Methods | RainDrop | |
| --- | --- | --- |
| | PSNR ↑ | SSIM ↑ |
| AttentiveGAN (Qian et al., 2018) | 31.59 | 0.9170 |
| DuRN (Liu et al., 2019) | 31.24 | 0.9259 |
| RaindropAttn (Quan et al., 2019) | 31.44 | 0.9263 |
| IDT (Xiao et al., 2022) | 31.87 | 0.9313 |
| RDDM (Liu et al., 2024) | 32.51 | **0.9563** |
| RainDropDiff (Özdenizci & Legenstein, 2023) | 32.29 | 0.9422 |
| + RestoreGrad (ours) | **32.69**±0.03 | 0.9441±7e-5 |

*Best values in bold and second best values underlined.

**Visualizing the Learned Prior:** We visualize the learned prior distribution variances for a chosen image input with various $\eta$ values in Figure 12 since we are interested in the effect of this newly introduced hyperparameter. We plot the results for the first channel of the image. The original contaminated image (i.e., the conditioner $\mathbf{y}$ to the DDPM model) is also presented for reference purposes. As expected for the latent space regularization effect, a large $\eta$ results in smaller variances as enforcing stronger regularization, while a small $\eta$ leads to larger variances, as observed in the plots. Moreover, the learned prior appears to preserve the structure of the image, indicating that it tends to learn a prior distribution that approximates the data distribution.

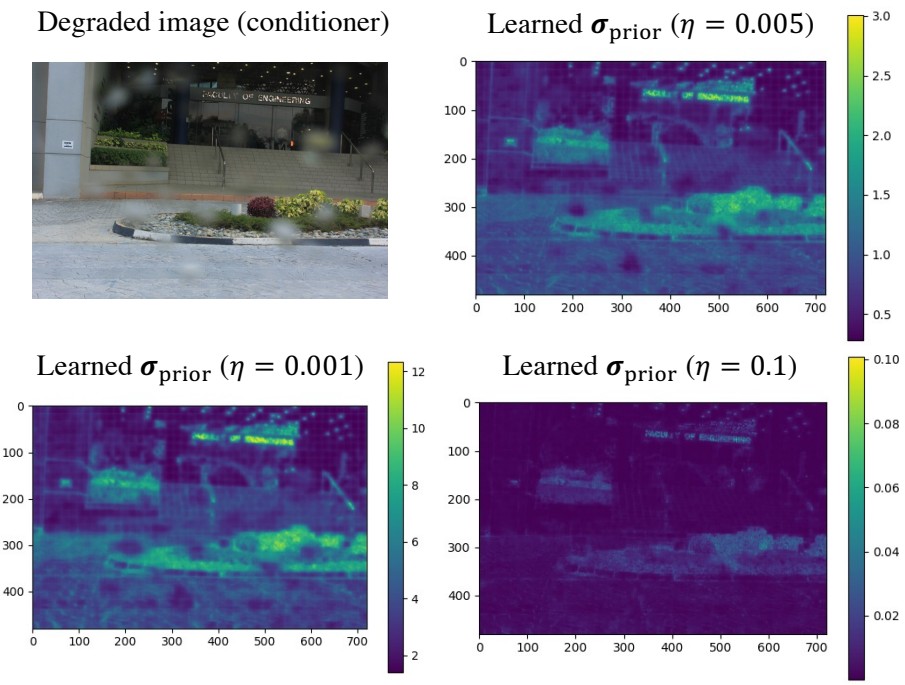

*Figure 12.* Visualization of learned prior distribution variances with various $\eta$ for a sample image taken from the RainDrop test set (Qian et al., 2018). Mind the magnitude color bar of each figure. We can see that a larger $\eta$ results in smaller variance of the prior distribution, while a smaller $\eta$ leads to larger variance.

**Restoration Performance vs. $\eta$ and $\lambda$:** We also study the IR performance of the RestoreGrad models trained across various combinations of $\eta$ and $\lambda$ in Table 11, where the models were trained and tested on the RainDrop dataset. The results show that RestoreGrad works effectively for a wide range of $\eta$ and $\lambda$ values to outperform the baseline DDPM model, RainDropDiff from Özdenizci & Legenstein (2023), which utilizes the standard Gaussian prior for the diffusion process.

*Table 11.* RestoreGrad performance for various $\eta$ and $\lambda$, where the models were trained on the RaindDrop training set (Qian et al., 2018) for 9,261 epochs and evaluated on the test set. The baseline RainDropDiff model scores reported in the original paper of Özdenizci & Legenstein (2023) (which was trained for 37,042 epochs, 4 times more than our RestoreGrad models) are also presented here for comparison purposes.

| Model | $\eta$ | $\lambda$ | PSNR ↑ | SSIM ↑ |
|---|---|---|---|---|
| RestoreGrad (ours) | 0.05 | 0.1 | **32.55** | **0.9440** |
| | 0.01 | | **32.73** | **0.9448** |
| | 0.005 | | **32.69** | **0.9441** |
| | 0.001 | | **32.63** | 0.9404 |
| | 0.0005 | | **32.50** | 0.9405 |
| RestoreGrad (ours) | 0.005 | 10 | **32.74** | **0.9442** |
| | | 1 | **32.72** | **0.9441** |
| | | 0.1 | **32.69** | **0.9441** |
| | | 0.01 | **32.41** | 0.9417 |
| RainDropDiff (Özdenizci & Legenstein, 2023) | - | - | 32.29 | 0.9422 |

*Values in bold text indicate better scores than the baseline ReainDropDiff model.

**Images Generated from Different Prior Noise Samples:** In Figure 13, we present example images generated for the same conditioner $\mathbf{y}$ using different random seed values (1, 10, and 20) when sampling the latent noise $\epsilon$, for both the baseline DDPM (RainDropDiff) and our RestoreGrad. In the figure, although it is challenging to perceive the difference between the results of different seed values simply by inspecting the images visually, the quality (PSNR and SSIM scores) of the images produced by RestoreGrad is consistently better than the baseline DDPM among the different seed values used, further demonstrating the superiority of our method in restoring higher fidelity signals.

**Signal Quality and Encoder Complexity Trade-Offs:** Similar to the computational complexity analysis provided in Table 2 for the SE task, we also present the results for the IR task on the RainDrop dataset in Table 12. From the table, we again observe that the performance of our approach slightly improves with the use of a larger encoder, while also with increasing complexity. However, the computational overhead of our approach, i.e., the additional complexity due to the encoder module(s), is again relatively smaller compared to the adopted DDPM module (i.e., RainDropDiff (Özdenizci & Legenstein, 2023)), in terms of processing latency (<1.3% of DDPM) and memory usage (<30% of DDPM).

*Table 12.* IR comparison of RestoreGrad models (on RainDrop dataset) using encoder modules of different sizes and the corresponding latency and GPU memory usage (measured on one NVIDIA Tesla V100 GPU) presented as the ratio of encoder to DDPM.

| Encoder size | PSNR↑ | SSIM↑ | Proc. Time | Memory |
|---|---|---|---|---|
| Base (0.27M) | 32.65 | 0.9414 | 0.9% | 16% |
| Large (1.9M) | 32.77 | 0.9444 | 1.3% | 30% |

**Visualization of Diffusion Processes:** We present a qualitative analysis of the diffusion processes by showing a sequence of images from $t = 0$ to $t = T$ for the baseline conditional diffusion model (WeatherDiff) and with the proposed method (RestoreGrad) in Figure 14. It can be seen that, at a given timestep $t$, the generated image by using RestoreGrad is cleaner than that of WeatherDiff, indicating a better diffusion trajectory developed by using our approach.

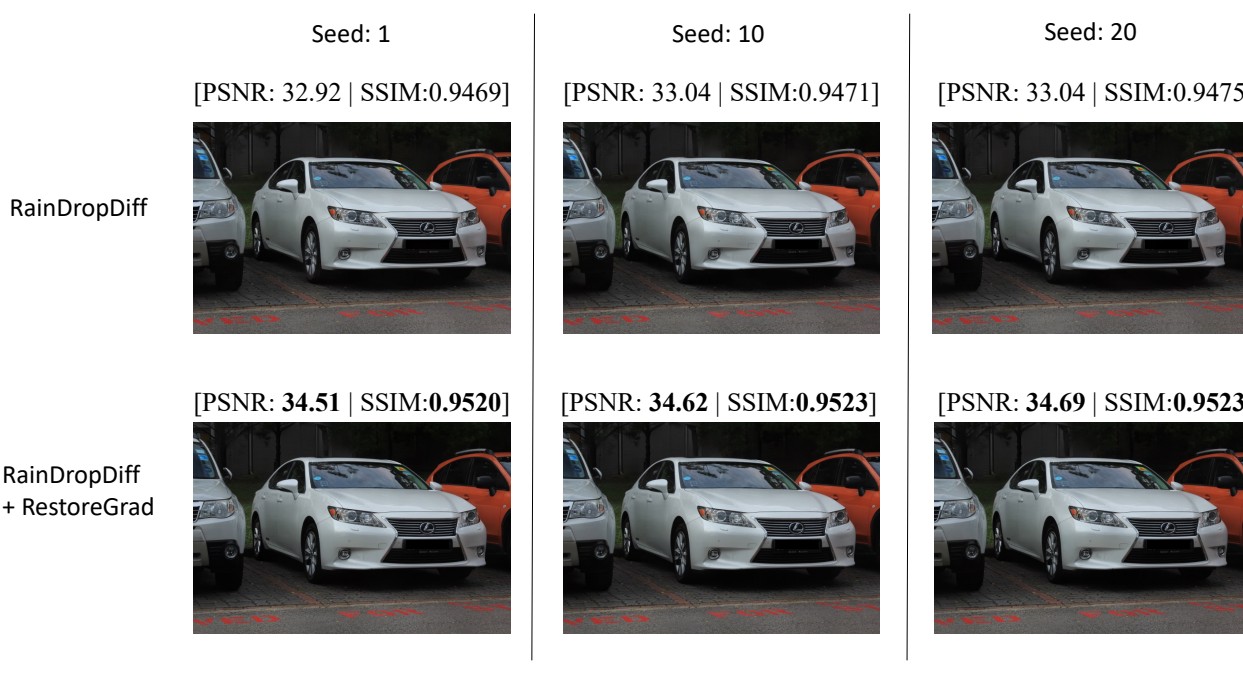

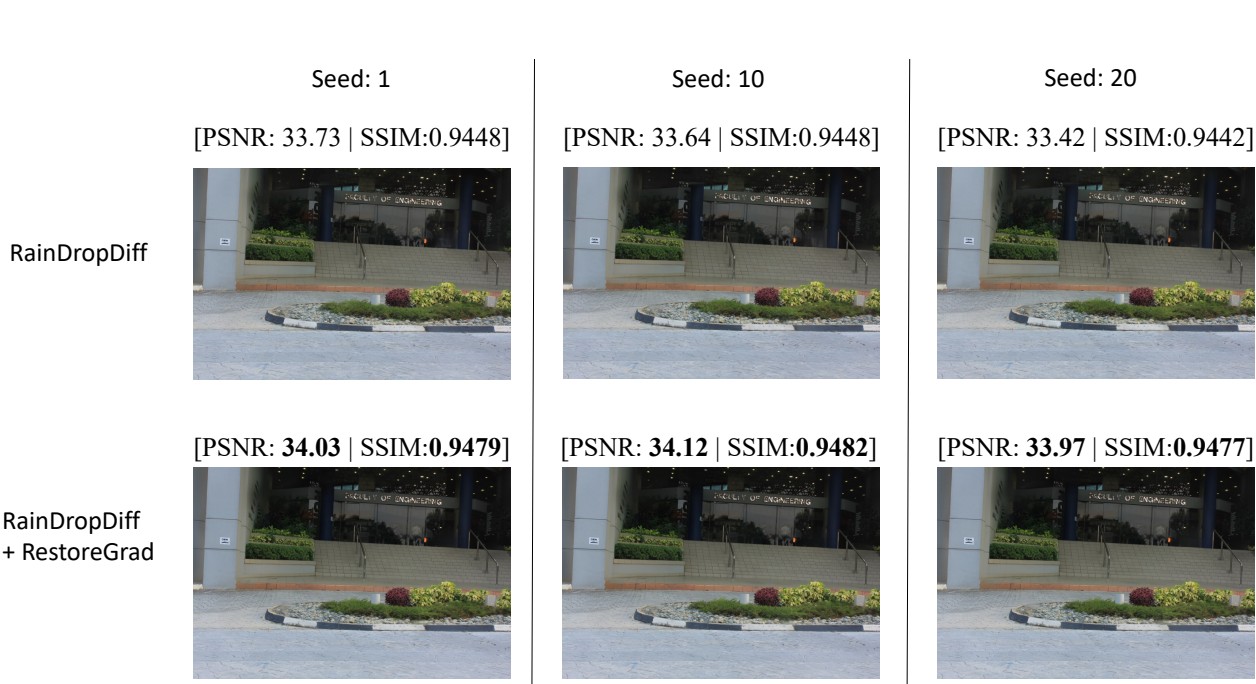

*Figure 13.* Example restored images from different prior noise samples (by using different random seed values) for the baseline DDPM (RainDropDiff) and our approach (RestoreGrad).

**forward process** ⟶ *t=T*

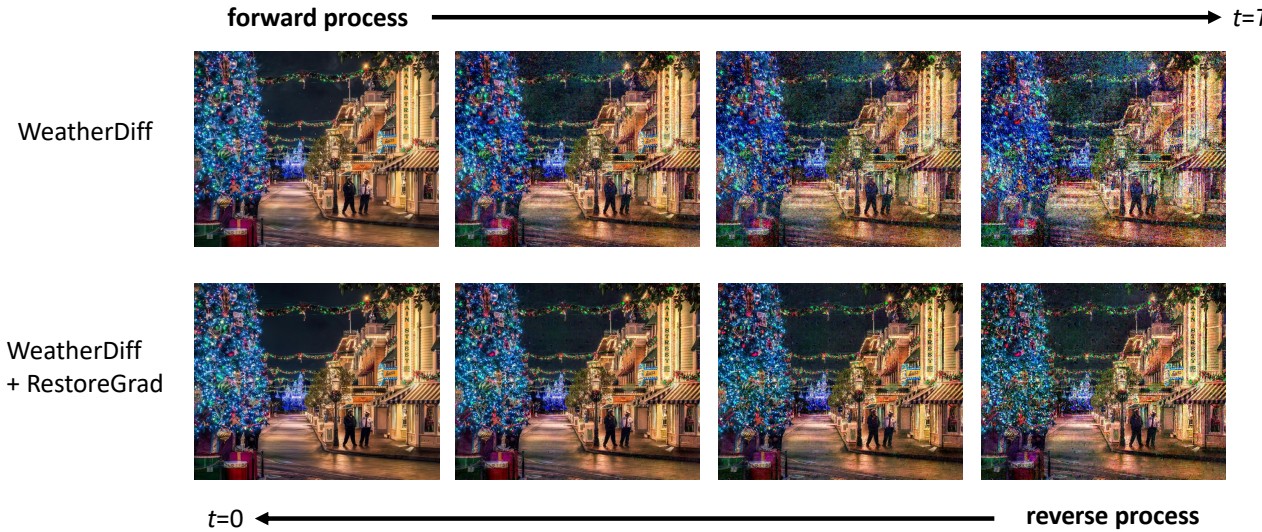

WeatherDiff

WeatherDiff
+ RestoreGrad

*t=0* ⟵ **reverse process**

*Figure 14.* Visualization of diffusion processes for the baseline DDPM (WeatherDiff) and our method (RestoreGrad).

**Experiments on Image Deblurring:** We apply RestoreGrad to the baseline conditional DDPM (cDDPM) which implements the same architecture as the patch-based DDPM of Özdenizci & Legenstein (2023) used for weather degradations for deblurring. We trained the baseline cDDPM and RestoreGrad models and validated their performance on the RealBlur dataset (Rim et al., 2020), a large-scale dataset of real-world blurred images captured both in the camera raw and JPEG formats, leading to two sub-datasets: *RealBlur-R* from the raw images and *RealBlur-J* from the JPEG images. Each training set consists of 3,758 image pairs and each test set consists of 980 image pairs. In Table 13, we present results of the baseline cDDPM and RestoreGrad models trained after 853 epochs. We also include scores of two existing models, SRN-DeblurNet (Tao et al., 2018) and DeblurGAN-v2 (Kupyn et al., 2019), which performed similarly to the baseline cDDPM (taken from results by Rim et al. (2020)), as references for comparison. We can see that, except for LPIPS and FID on RealBlur-J, RestoreGrad is able to achieve improved scores than the baseline cDDPM, and outperform the compared methods.

*Table 13.* Image deblurring of realistic blurred images.

| Methods | RealBlur-J | | | | RealBlur-R | | | |
|---|---|---|---|---|---|---|---|---|
| | PSNR↑ | SSIM↑ | LPIPS↓ | FID↓ | PSNR↑ | SSIM↑ | LPIPS↓ | FID↓ |
| SRN-DeblurNet | 31.38 | 0.9091 | - | - | 38.65 | 0.9652 | - | - |
| DeblurGAN-v2 | 29.69 | 0.8703 | - | - | 36.44 | 0.9347 | - | - |
| Baseline cDDPM | 30.69 | 0.9043 | **0.220** | **15.17** | 37.71 | 0.9777 | 0.126 | 14.46 |
| + RestoreGrad (ours) | **31.51** | **0.9095** | 0.224 | 15.53 | **38.78** | **0.9796** | **0.122** | **13.61** |

*Bold text for best and underlined text for second best values.

**Experiments on Image Super-Resolution:** We further study the benefits of RestoreGrad over the baseline conditional DDPM (cDDPM) model on image super-resolution tasks with the DIV2K dataset (Agustsson & Timofte, 2017; Timofte et al., 2017). We compare RestoreGrad with the baseline cDDPM model (the same architecture of the patch-based DDPM of WeatherDiff (Özdenizci & Legenstein, 2023)) for ×2 and ×4 downscale factor subsets (with bicubic downgrading operators). There are 800 images for training and 100 images for validation in each subset. For both subsets, we trained a baseline cDDPM and the RestoreGrad models for 2000 epochs on the training set and evaluated their performance on the corresponding validation set. The results are presented in Table 14, where we can see that except for the LPIPS metric, RestoreGrad is more beneficial then the baseline cDDPM in terms of achieving better scores in the other three metrics.

*Table 14.* Comparison of baseline conditional DDPM (cDDPM) and the RestoreGrad on image super-resolution tasks.

| Methods | DIV2K ×2 | | | | DIV2K ×4 | | | |
|---|---|---|---|---|---|---|---|---|
| | PSNR ↑ | SSIM ↑ | LPIPS ↓ | FID ↓ | PSNR ↑ | SSIM ↑ | LPIPS ↓ | FID ↓ |
| Baseline cDDPM | 27.40 | 0.9291 | **0.127** | 7.577 | 25.18 | 0.8064 | **0.269** | 7.849 |
| + RestoreGrad (ours) | **27.56** | **0.9341** | 0.136 | **7.547** | **25.56** | **0.8228** | 0.290 | **7.839** |

*Better values are indicated with bold text.

**More Image Restoration Examples:** We provide more examples in Figures 15, 16, 17, 18, 19, for comparing our RestoreGrad with the baseline DDPM approaches (i.e., RainDropDiff, WeatherDiff) of Özdenizci & Legenstein (2023). The restored image of RainDropDiff in Figure 15 was obtained by using the model weights trained by ourselves. The restored images of WeatherDiff in Figures 16, 17, 18, 19 were obtained by using the trained model weights provided by Özdenizci & Legenstein (2023) at `https://github.com/IGITUGraz/WeatherDiffusion`. We also provide examples of image deblurring in Figures 20 and 21.

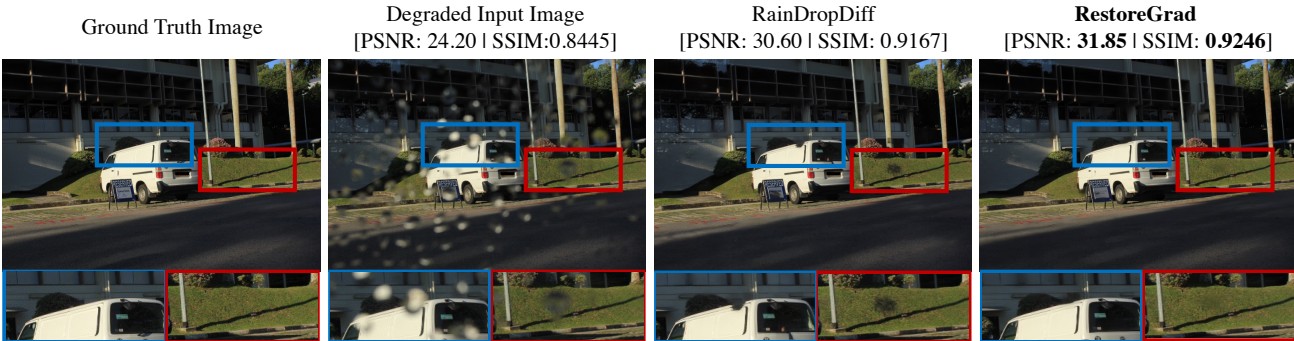

*Figure 15.* Restored images by RainDropDiff (Özdenizci & Legenstein, 2023) and RestoreGrad (ours) for a test sample from the RainDrop test set.

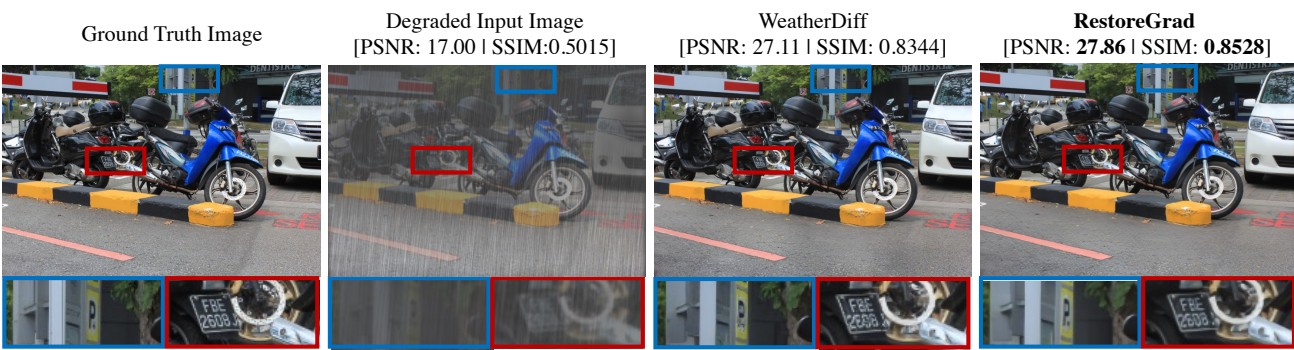

*Figure 16.* Restored images by WeatherDiff (Özdenizci & Legenstein, 2023) and RestoreGrad (ours) for a test sample from the Outdoor-Rain test set.

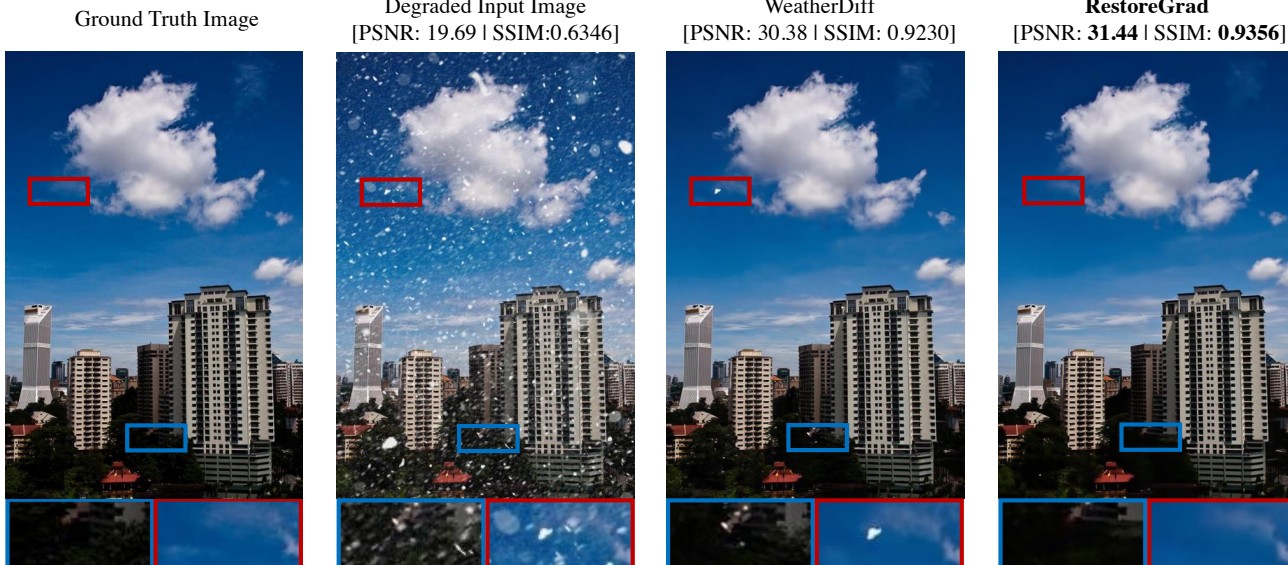

*Figure 17.* Restored images by WeatherDiff (Özdenizci & Legenstein, 2023) and RestoreGrad (ours) for a test sample from the Snow100K-L test set.

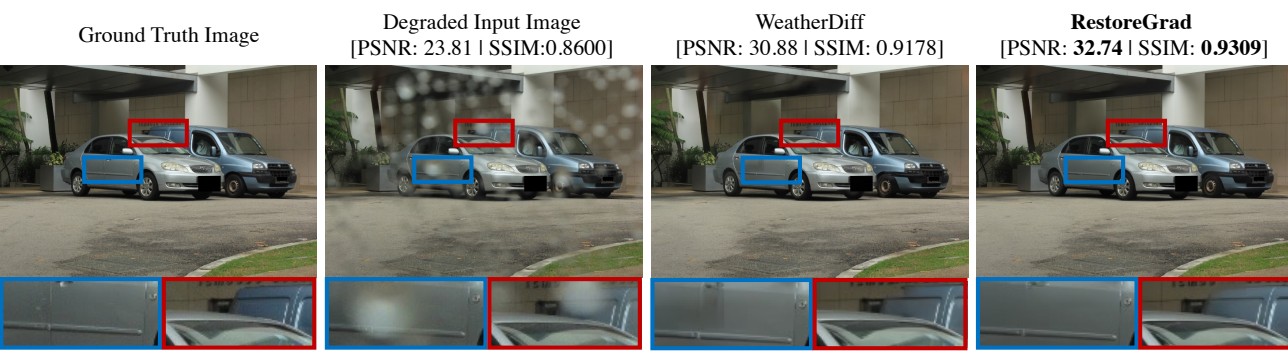

*Figure 18.* Restored images by WeatherDiff (Özdenizci & Legenstein, 2023) and RestoreGrad (ours) for a test sample from the RainDrop test set.

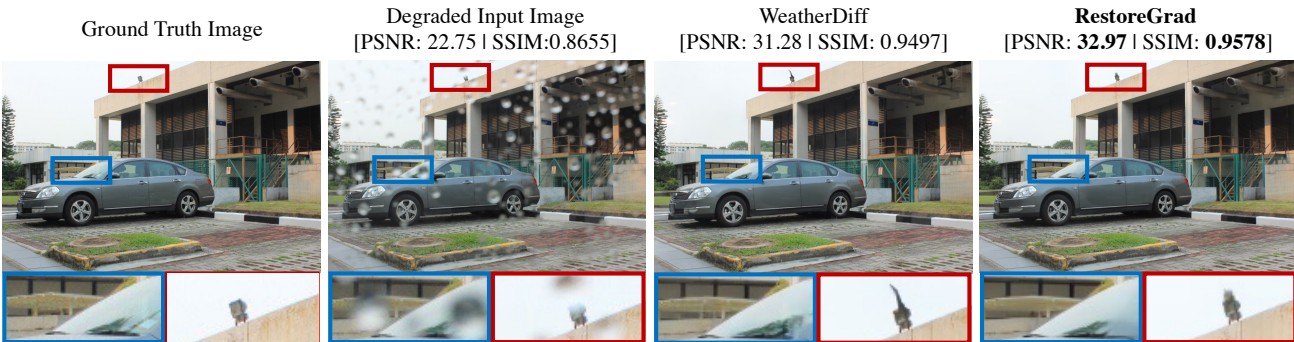

*Figure 19.* Restored images by WeatherDiff (Özdenizci & Legenstein, 2023) and RestoreGrad (ours) for a test sample from the RainDrop test set.

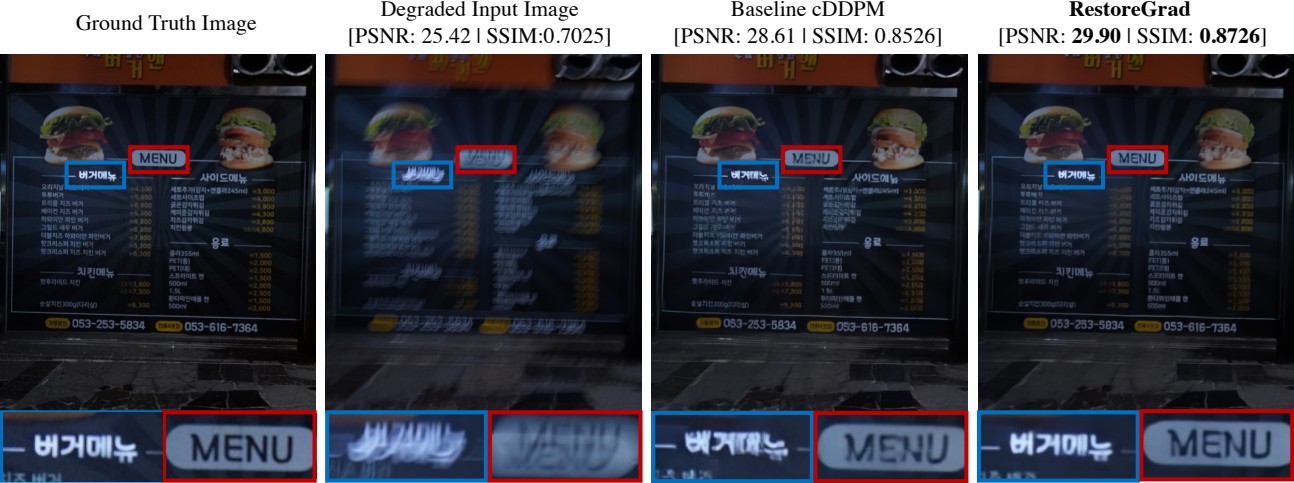

*Figure 20.* Image deblurring examples using a test image taken from the RealBlur test set.

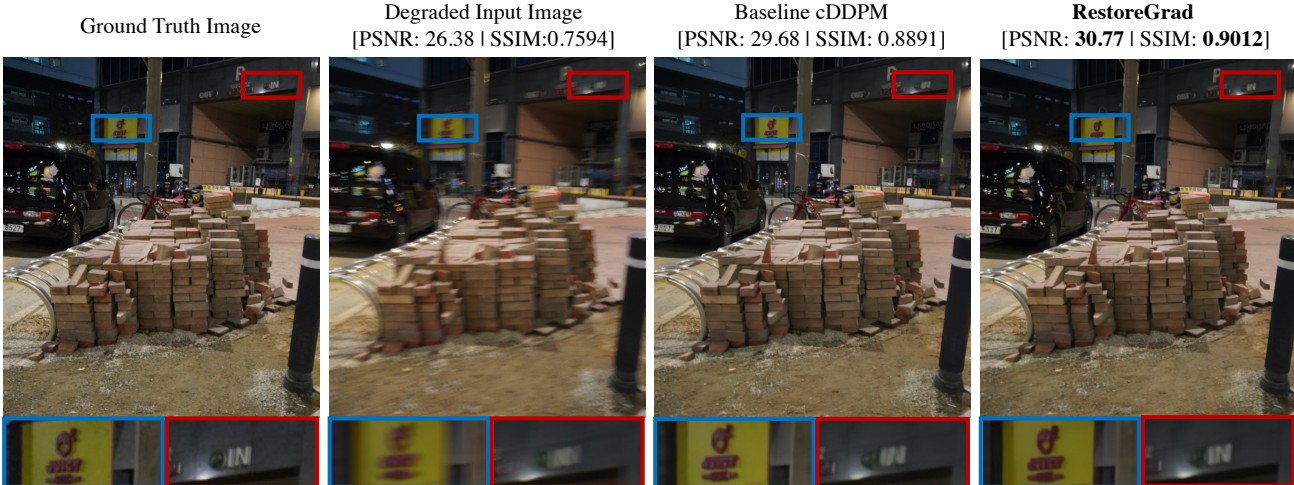

*Figure 21.* Image deblurring examples using a test image taken from the RealBlur test set.

