# OpenReview forum: "RestoreGrad: Signal Restoration Using Conditional Denoising Diffusion Models with Jointly Learned Prior"
_ICML.cc/2025/Conference — ICML 2025 poster_

### Official Review · Reviewer_JrVk · 2025-03-11

**Overall Recommendation:** 4

**Summary:**

The authors present a new type of conditional diffusion model with an application image restoration.
An image $x_0$ is predicted conditioned on a degraded version $y$.
The presented method is based on interpreting the diffusion model as a variational autoencoder (VAE). Based on this idea, they allow the prior over the noise $\epsilon$ at $t=T$ to be conditioned on $y$ (implemented by a CNN that outputs per-pixel standard deviations).
Additionally, they model the posterior $q(\epsilon|x_0, y)$ as conditioned on $y$.
The authors apply their method to image and audio data and show that it produces superior results for especially with a reduced number of iterations during inference and allows for faster convergence during training.

## update after rebuttal

I appreciate the detailed response by the authors and will stick with my original 'accept' rating.

**Claims And Evidence:**

**"We study the problem of learning the prior distribution jointly with the conditional DDPM for signal restoration applications, aiming at providing a more systematic, learning-based treatment to address the inefficiency incurred by existing selections of the prior distribution."**
This is correct, they paper looks at this problem and finds a sensible solution.

**"We propose a new framework called RestoreGrad that learns the prior in conjuncture with the DDPM model through a prior encoder"**
This is also correct. The method is clearly described.

**Performance claims:**

**"faster convergence (5-10 times fewer training steps)"**
This claim is experimentally validated.

**"improved robustness to using fewer sampling steps in inference time (2-2.5 times fewer)"**
This claim is experimentally validated

**Essential References Not Discussed:**

I did not notice any missing references to related literature.

**Experimental Designs Or Analyses:**

I agree with the experimental design and especially appreciate the authors checking the effect of $\nu$ and the importance of training the posterior net.
I wonder how well the method would preform if the posterior net would only be conditioned on $x_0$ and not on $y$.
This could even be an interesting experiment in a non conditional setting. could a non conditional diffusion model benefit from using a posterior net conditioned on  $x_0$?

**Methods And Evaluation Criteria:**

I believe this is an elegant and sensible approach. The methodology follows directly form viewing the diffusion model as a VAE.
I think the evaluation datasets and baselines make a lot of sense and are convincing.

However, I think the qualitative evaluation and discussion could be improved and makes me a bit worried:

* The learned prior in Figure 3 shows grid patterns, especially visible in the top right corner, even though there are no grid structures in the image. These are also visible in supplementary Figure 12. The authors do not comment on this grid pattern. It would be interesting to see examples of learned priors for other images and to hear a discussion on why these artefacts occur.

* I am missing a qualitative analysis of the generation process. It would be great to see a sequence of images showing the generation form t=T to t=0 for the proposed method and a vanilla conditional diffusion model.

* I wonder of the learned prior effects the diversity of generated images. This could be qualitatively checked by showing multiple images generated for the same $y$ for the proposed method and a vanilla baseline.

**Other Comments Or Suggestions:**

**Parametrization of prior and posterior net:**
Is there a reason why the prior and posterior net predict only the standard deviations for each pixel and not the mean?
Would this not make the method potentially more powerful?

**Typos:**

"To avoid intractable integral" -> "To avoid the intractable integral"

**Other Strengths And Weaknesses:**

I don't have additional points beyond the ones made above.

**Questions For Authors:**

I don't have additional questions beyond the ones made above.

**Relation To Broader Scientific Literature:**

The authors apply a VAE perspective to conditional diffusion models, which enables them to apply naturally concepts from literature on conditional VAEs to diffusion models, yielding a way to train a prior and posterior together with the normal diffusion unet.

**Theoretical Claims:**

I checked the equations in the main paper and they seem valid as far as I can tell.
I am not completely sure, I a agree with the reasons the authors give for why their method gives improved results.
For example they say:
*"However, existing adoption of the standard Gaussian as the prior distribution in turn discards such information* [form $y$] *, resulting in sub-optimal performance."*
At each step of the generation with a vanilla conditional diffusion model, y is provided as input, so all information is available (albeit not in the noisy image but in the ) and no information is disregarded.

Looking at this from the perspective of conditional VAEs, I think the proposed method corresponds to Figure 2 in [1] versus the vanilla conditional diffusion model that corresponds to Figure 1 in [1].
These are two different types of models and it's not obvious for me form a theoretical perspective why one is to be preferred over the other.
Can the authors comment on this?





[1]: https://beckham.nz/2023/04/27/conditional-vaes.html#sec_zy_independent

---

> ### Author Rebuttal · Authors · 2025-03-31
>
> We greatly appreciate that the reviewer found our method elegant and sensible, and that our experimental validation makes sense and is convincing. Below, we address the main concerns and questions.
>
> ---
>
> ## Methods And Evaluation Criteria
>
> > **The learned ...**
>
> Thank you for carefully checking our Figures 3 and 12 and observing the details (grid patterns). Please note that the figures are not the restored images, **but a visualization of the learned priors**. The purpose of visualizing the priors is to demonstrate that they are informative and structured. In terms of the grid patters, we point out that the priors are actually estimated in a **patch-based manner**, following the practice of our conditional DDPM backbone, WeatherDiff (Özdenizci \& Legenstein, 2023). As a result, the estimated priors might present the grid patterns between patches due to potential discontinuity. However, this is alleviated in the later stage when merging the restored image patches through the *mean estimated noise based sampling updates for the overlapping pixels* used in WeatherDiff. Therefore, the final restored images (our Figures 7, 13-19) are free from grid structures.
>
> > **I am missing ...**
>
> Please refer to the figure via the anonymous [Link](https://anonymous.4open.science/r/restoregrad-8A09/diffusion_process.png).
>
> > **I wonder ...**
>
> The applications considered in this work, i.e., restoring a signal from degraded observations, prioritize the output signal quality over diversity. Specifically, the models are trained to generate signals close to the original clean signal $\mathbf{x}_0$ given the conditioner $\mathbf{y}$, instead of producing diverse signals.
>
> ---
>
> ## Theoretical Claims
>
> > **I checked ...**
>
> For the claim: ``However, existing adoption ...'', we meant to say that **in the stage of estimating the prior distribution** that the information from $\mathbf{y}$ is discarded. We will make sure to rectify the claim for accuracy.
>
> > **Looking at ...**
>
> We thank the reviewer for providing the interesting connection between our approach and the conditional VAE shared in the blog post. Our reason for why the dependent scheme may be preferred over the independent scheme is that, a data-dependent prior can potentially improve the diffusion trajectory of DDPMs in signal restoration. Specifically, as we can exploit the information from $\mathbf{y}$ which adequately correlates with $\mathbf{x}_0$, it is possible to form a prior distribution that is closer to the target distribution, improving the trajectory of the diffusion process for more efficient training and inference.
>
> ----
>
> ## Experimental Designs Or Analyses
>
> > **I agree ...**
>
> As shown in the table below, the RestoreGrad model is still able to improve over the baseline DDPM (CDiffuSE) considerably when Posterior Net is conditioned only on $\mathbf{x}_0$. This makes sense, as the information of $\mathbf{x}_0$ is still available to the model during training which is important for estimating a desirable prior.
>
> | SE model                                              | PESQ $\uparrow$ | COVL $\uparrow$ | SSNR $\uparrow$ | SI-SNR $\uparrow$ |
> |-------------------------------------------------------|-----------------|-----------------|-----------------|-------------------|
> | CDiffuSE (96 epochs)                                  | 2.32            | 2.89            | 3.94            | 11.84             |
> | + RestoreGrad                                         | 2.51            | 3.14            | 5.92            | 14.74             |
> | + RestoreGrad (only $\mathbf{x}_0$ for Posterior Net) | 2.52            | 3.13            | 5.58            | 14.48             |
>
> Our framework concentrates on signal restoration to generate a signal as close to the original signal as possible. In terms of the unconditional setting, if the task emphasizes more on diversity of generated data, then the data-dependent prior may not benefit that much. Unfortunately, our current work have not explored the unconditional setting and we do not have a concrete answer, but we sincerely hope that our work can be a good starting point for the research going forward.
>
> ---
>
> ## Questions For Authors
>
> > **Parametrization ...**
>
> In theory we can also parameterize the prior and posterior encoders to predict the mean in addition to the (co)variance, which may in turn require more capacity of the encoders to estimate both entities adequately. In this work, we have chosen to demonstrate the idea of learnable priors by estimating only the variance for simplicity. Our reason for that is, as the mean term does not contribute to the randomness of the samples (which only shifts the distribution), we reserve the capacity of the encoder modules for estimating the variance term that actually controls the shape (spread) of the distribution for sampling. In signal restoration applications, we also exploit normalization of the signals to zero mean and perform sampling in the normalized domain for efficiency.

---

> > ### Comment · Reviewer_JrVk · 2025-04-08
> >
> > Thank you for your detailed response!
> > A quick follow-up-question: The authors write:
> > "The applications considered in this work, i.e., restoring a signal from degraded observations, prioritize the output signal quality over diversity. Specifically, the models are trained to generate signals close to the original clean signal given the conditioner , instead of producing diverse signals."
> > Still, this is an ill-posed problem we are addressing, so there should be a posterior distribution of possible solutions. This is the reason, why generative models produce superior outputs versus simple regression techniques. Would the authors include a Figure with multiple samples in their supplement in order to visualise the diversity of the output?

---

> > > ### Author Response · Authors · 2025-04-09
> > >
> > > We sincerely thank the reviewer for the follow-up question. Yes, we agree with the reviewer that the problem considered (signal restoration) is generally ill-posed, and that generative models achieve better performance over their deterministic counterparts by estimating the data distribution to generate possible solutions. With that, one of the advantages of generative models is that they are typically more robust to unseen conditions (e.g., out-of-distribution (OOD) samples) than deterministic models, resulting from their abilities to generate realistic data from the learned distribution. We would like to take this opportunity to highlight that, as evidence of RestoreGrad's improved robustness, we have provided the comparison on OOD data for the SE task (in Table 6 of the submitted manuscript), showing that RestoreGrad stands out among other deterministic (Demucs, WaveCRN) and generative (DOSE) models. For visualization of the generated outputs, following the reviewer's suggestion, we further provide multiple images generated for the same $\mathbf{y}$ using the RestoreGrad and DDPM baseline (i.e., RainDropDiff) via the anonymous [Link](https://anonymous.4open.science/r/restoregrad-8A09/images_different_seeds.png). There, we present restored images using different random seeds when sampling the latent noise. In the figure, we note that it is challenging to perceive the difference between the results of different seeds simply via inspecting the images visually, for both the baseline DDPM and RestoreGrad. However, as can be seen in the figure, the quality (PSNR and SSIM scores) of the images produced by RestoreGrad is consistently better than the baseline DDPM among the different seeds used, demonstrating the superiority of our method in restoring higher fidelity signals. Again, we thank the reviewer for bringing up the question; we will include the above discussion in our revised paper for completeness.

---

### Official Review · Reviewer_wfM7 · 2025-03-12

**Overall Recommendation:** 3

**Summary:**

The paper proposes a novel framework to enhance the efficiency of conditional Denoising Diffusion Probabilistic Models (DDPMs) for signal restoration tasks, such as speech enhancement (SE) and image restoration (IR). By jointly learning a data-informed prior distribution using two encoder networks (Prior Net and Posterior Net) alongside the DDPM, the method aims to address the inefficiency of traditional DDPMs caused by the mismatch between the standard Gaussian prior and real data distributions. The authors integrate DDPM into a variational autoencoder (VAE) framework, deriving a new evidence lower bound (ELBO) to optimize the model. Extensive experiments on benchmark datasets demonstrate improved convergence speed, signal quality, and robustness compared to baseline DDPMs and PriorGrad.

**Claims And Evidence:**

Yes.

**Essential References Not Discussed:**

The authors should provide a comparison section to discuss the work of image restoration in detail, because there are already a lot of works in this category, otherwise, the content of this section is insufficient.

**Experimental Designs Or Analyses:**

1. The paper claims image restoration. It also compares many different tasks, but I am curious about the results of image denoising. Can the author make further comparisons? In addition, the baseline selected for comparison seems to be insufficient and a bit old.

2. The methods compared in the article seem to be very old algorithms, such as WeatherDiffusion, which seems to be a method from 23 years ago. The latest CVPR'24 and ECCV'24 also have newer algorithms. The author must compare them to prove the superiority of the methods and their similarities and differences.

**Methods And Evaluation Criteria:**

Almost correct.

**Other Comments Or Suggestions:**

Please refer to the weaknesses.

**Other Strengths And Weaknesses:**

Strengths:

1. The proposal of systematically learning the prior distribution for DDPMs, rather than relying on handcrafted priors, is a significant contribution that addresses a key limitation in prior work (e.g., PriorGrad). This fills a gap in improving DDPM efficiency in a domain-agnostic manner.

2. Embedding DDPM into a VAE framework with a novel ELBO formulation is theoretically sound and creative, effectively combining the generative power of DDPMs with the modeling efficiency of VAEs.

3. The method’s ability to reduce training time (e.g., 10× faster convergence in SE) and maintain quality with fewer inference steps enhances its applicability in real-world scenarios.

4. The article compares multiple data sets and can almost prove the effectiveness and robustness of the method.
Weaknesses:

1. The assumption that the final diffusion step $\mathbf{x}_T$ equals the latent variable $\epsilon$ (Proposition 3.1) is critical but lacks rigorous justification. Its validity may depend on the number of diffusion steps $T$ and the noise schedule $\beta_t$, which are not thoroughly analyzed.

2. The fixed covariance assumption ($\sigma_t^2 = \hat{\beta}_t$) in the reverse process simplifies training but may limit the model’s flexibility in capturing complex data distributions. Alternative approaches are not explored.

3. The paper claims image restoration. It also compares many different tasks, but I am curious about the results of image denoising. Can the author make further comparisons? In addition, the baseline selected for comparison seems to be insufficient and a bit old.

4. The methods compared in the article seem to be very old algorithms, such as WeatherDiffusion, which seems to be a method from 23 years ago. The latest CVPR'24 and ECCV'24 also have SOTA diffusion-based algorithms. The author must compare them to prove the superiority of the methods and their similarities and differences.

5. More comparisons on computational overhead are needed. Please provide more details on this part.

6. The authors should further analyze the limitations of this method and the error cases, and explain why.

**Questions For Authors:**

Please refer to the weaknesses.

**Relation To Broader Scientific Literature:**

It advances the field of generative modeling by enhancing the efficiency of DDPMs. It introduces a systematic, learning-based prior, aligning with trends in data-driven generative models and hybrid frameworks.

**Theoretical Claims:**

Almost correct.

---

> ### Author Rebuttal · Authors · 2025-03-31
>
> We thank the reviewer for acknowledging that our method is a significant contribution to address key limitation in existing works, and is theoretically sound and creative. In the following, we provide our responses to address the main questions and concerns.
>
> ---
>
> ## Experimental Designs Or Analyses
>
> Our framework aims to provide a general technique to improve conditional DDPM based Signal Restoration approaches. We have provided evidence of improved performance across multiple signal restoration tasks. We believe the capabilities and generality of our approach have the potential to extend to other applications like image denoising, and even to other signal modalities.
>
> We would like to point out that the baseline image restoration model we used, i.e., WeatherDiff (Özdenizci and Legenstein, 2023), was published in 2023 and considered a reasonably strong baseline which achieved SOTA performance in several restoration tasks. Also, our main goal is to show the feasibility of learnable priors in a generic setting, rather than pursuing the SOTA performance in a specific task. However, as the reviewer suggested, it is still a good idea to compare with the latest methods, e.g., Diffusion Ye et al., (2024) and Liu et al., (2024). We have further provided such comparison in Tables A and B through the anonymous [Link](https://anonymous.4open.science/r/restoregrad-8A09/comparison.pdf).
>
> **Refs:**
>
> - Özdenizci and Legenstein. Restoring vision in adverse weather conditions with patch-based denoising diffusion models. *IEEE TPAMI*, 2023.
> - Ye et al. Learning diffusion texture priors for image restoration. In *CVPR*, 2024.
> - Liu et al. Residual denoising diffusion models. In *CVPR*, 2024.
>
> ---
>
> ## Essential References Not Discussed
>
> As the reviewer mentioned, a significant amount of research has been conducted in the field of IR, and we might have missed some of the important ones. We will include more discussion on recent diffusion-based approaches and refer the reader to latest survey papers, e.g., He et al. (2025). As this is a rapidly developing field with many emerging approaches, we may not be able to discuss them all. If the reviewer has any specific works in mind that are important but we have missed, please kindly let us know.
>
> **Ref:**
>
> - He et al. Diffusion models in low-level vision: A survey. *IEEE TPAMI*, 2025.
>
> ---
>
> ## Other Strengths And Weaknesses
>
> ### Weaknesses
>
> > **No. 5**
>
> In fact, the assumption is used in the standard DDPM by assuming a large enough $T$ and a carefully designed variance schedule $\\{\\beta\_t\\}\_{t=1}^T$. Thus, it is not new here and has been adopted in DDPM literature to sample $\mathbf{x}_T$ from the latent variable space of $\boldsymbol{\epsilon}$. What is new in our approach is that we explore the feasibility of having the distribution of $\boldsymbol{\epsilon}$ learnable. We will explicitly mention the conditions to improve the rigor of Proposition 3.1.
>
> > **No. 6**
>
> We have followed the common practice in the literature to model the reverse process with a fixed variance schedule. In the future, it is also possible to combine our method with alternative approaches, such as Improved DDPM (Nichol \& Dhariwal, 2021) which learns the variances instead of using a fixed noise schedule, to further improve the modeling efficiency.
>
> **Ref:**
>
> - Nichol and Dhariwal. Improved denoising diffusion probabilistic models. In *ICML*, 2021.
>
> > **No. 7 and No. 8**
>
> Already responded in Experimental Designs Or Analyses.
>
> > **No. 9**
>
> From Table 2 of our submitted paper, we observed that the computational overhead due to the encoder module is relatively small compared to the DDPM module, in terms of processing latency (2\% of DDPM) and memory usage (10\% of DDPM). We have further conducted similar experiment on the IR task (on the RainDrop dataset) to measure the latency and GPU memory usage (presented as the ratio of encoder to DDPM) shown in the table below:
>
> | Encoder size | PSNR$\uparrow$ | SSIM$\uparrow$ | Proc. Time | Memory |
> |--------------|----------------|----------------|------------|--------|
> | Base (0.27M) | 32.65          | 0.9414         | 0.9\%      | 16\%   |
> | Large (1.9M) | 32.77          | 0.9444         | 1.3\%      | 30\%   |
>
> > **No. 10**
>
> Our method is quite general in the sense that it can potentially be adopted in existing conditional DDPM models while introducing minimal additional complexity. However,  there are definitely limitations. First, we have focused on signal restoration where we suitably assume a zero-mean Gaussian prior distribution. Second, our approach improves the conditional DDPM by exploiting the decent correlation between the conditioner signal $\mathbf{y}$ and the target signal $\mathbf{x}_0$. However, if such correlation is rather weak or very unstructured, our current encoding scheme may not be able to directly extract useful information from there. In this case, a more sophisticated encoding strategy may be required.

---

### Official Review · Reviewer_2bPR · 2025-03-15

**Overall Recommendation:** 3

**Summary:**

This paper proposes RestoreGrad to restore signals with diffusion models. In order to achieve this, the paper proposes to train a prior net and a posterior net jointly with the diffusion model so that the noise can be sampled from the noise distribution output by the prior net. During training, three losses are applied to optimize the three networks: the latent regularization loss, the prior matching loss and denoising matching loss which is the original diffusion loss. During inference, the posterior net is removed and the prior net is used to output the distribution that is conditioned on the input so that the noise could be sampled directly from this distribution. The experiments are performed mainly on speech enhancement and image restoration.

**Claims And Evidence:**

Yes.

**Essential References Not Discussed:**

N/A

**Experimental Designs Or Analyses:**

The soundness/validity of the experimental designs looks good.

**Methods And Evaluation Criteria:**

Yes.

**Other Comments Or Suggestions:**

N/A

**Other Strengths And Weaknesses:**

Strengths
1. The paper is clearly written and the proposed method is interesting by utilizing the posterior net and the prior net.
2. The quantitative results look competitive.

Weaknesses
1. The proposed method seems incremental compared with PriorGrad. The key idea is similar, with the main difference being the jointly trained prior and posterior encoders.
2. The model requires the joint training of prior encoder and posterior encoder, which might be costly if the model or the input signal is large.

**Questions For Authors:**

I'm wondering if the method could be utilized for more state-of-the-art and more foundational image diffusion models e.g., SDXL? In this way.  a much powerful image distribution could be utilized.

**Relation To Broader Scientific Literature:**

The idea of learned prior is well studied in previous works like PriorGrad. The key idea here is to match the prior distribution with the posterior distribution so that during inference, the learned condition prior distribution could be used to sample the noise.

**Theoretical Claims:**

I checked the proof of Propositions 3.1 and 3.2 in the appendix roughly and they appear good to me.

---

> ### Author Rebuttal · Authors · 2025-03-31
>
> We greatly appreciate that the reviewer found our method interesting and the quantitative results competitive. In this rebuttal, we provide our responses to address the main concerns and questions.
>
> ---
>
> ## Relation To Broader Scientific Literature
>
> > **The idea ...**
>
> We would like to take this opportunity to highlight that, *the authors of PriorGrad were not able to demonstrate benefits of using learning-based priors, and they eventually just settled on using a handcrafted prior*. In other words, *the idea of learned priors was not yet well studied in the work of PriorGrad*. In contrast, our framework successfully demonstrates the **feasibility of leveraging learnable priors** for improving the efficiency of conditional DDPMs, including studies on practical nuances that make our proposed idea work in practice, which PriorGrad failed to demonstrate.
>
> ---
>
> ## Other Strengths And Weaknesses
>
> ### Weaknesses
>
> > **The proposed ...**
>
> We sincerely thank the reviewer for mentioning the comparison of our method with PriorGrad. We further clarify their differences.
>
> 1. PriorGrad only utilizes information from the condition signal $\mathbf{y}$ to compute the data-dependent prior. **As such, PriorGrad is not optimal for the applications where the condition signal contains certain degradations, e.g., in speech enhancement where $\mathbf{y}$ is a noisy version of the target $\mathbf{x}\_0$**. In contrast, *our RestoreGrad exploits information from the target signal $\mathbf{x}\_0$ in addition to $\mathbf{y}$ when learning to predict the prior.*
>
> 3. We have introduced *a new loss function for jointly learning priors by employing the prior and posterior encoders, $\psi$ and $\phi$*. Let us compare ours with PriorGrad:
>
> - PriorGrad:
> $$
> \min\_{\theta} \\,\\,\\, ||\boldsymbol{\epsilon}-\boldsymbol{\epsilon}\_\theta(\mathbf{x}\_t,\mathbf{y},t)||^2\_{\boldsymbol{\Sigma}^{-1}\_{y}}, \\,\\,\\, \text{where} \\,\\,\\, \boldsymbol{\Sigma}\_{y}=f(\mathbf{y}) \\, \text{and} \\, f(\cdot) \\, \text{is a pre-defined function that maps} \\, \mathbf{y} \\, \text{into the prior distribution.}
> $$
>
> - RestoreGrad (ours):
> $$
> \min_{\theta,\phi,\psi} \\,\\,\\, \eta\bigr(\bar{\alpha}\_T||\mathbf{x}\_0||^2\_{\boldsymbol{\Sigma}^{-1}\_{\text{post}}}+\log|\boldsymbol{\Sigma}\_{\text{post}}|\bigr)+||\boldsymbol{\epsilon}-\boldsymbol{\epsilon}\_\theta(\mathbf{x}\_t,\mathbf{y},t)||^2\_{\boldsymbol{\Sigma}^{-1}\_{\text{post}}}+\lambda \bigr(\log\frac{|\boldsymbol{\Sigma}\_{\text{prior}}|}{|\boldsymbol{\Sigma}\_{\text{post}}|}+\text{tr}(\boldsymbol{\Sigma}\_{\text{prior}}^{-1}\boldsymbol{\Sigma}\_{\text{post}})\bigr).
> $$
>
> We can see that our framework takes care of the prior estimation by utilizing encoders, by-passing the requirement to search for a suitable mapping function $f(\cdot)$ requiring the domain knowledge of a specific task (e.g., spectral analysis in speech tasks). Our framework is thus **more general and applicable to more modalities** as we demonstrated.
>
> > **The model ...**
>
> In this work, we have shown that our framework achieves significant improvements by using *standard*, *lightweight* architectures (ResNets) for the Prior and Posterior Nets. Specifically, the encoder modules we used were only 2\% and 0.3\% of the DDPM models in size for the SE and IR tasks, respectively. Such small encoders have already led to considerable performance boost for the DDPM backbone, and the computational overhead for the encoders is relatively small compared to the DDPM itself (e.g., see Table 2 of the submitted manuscript and new results for IR at [Link](https://openreview.net/forum?id=NbjrGgxLPi&noteId=Dc1KIYzvCF)).
>
> ---
>
> ## Questions For Authors
>
> > **I'm wondering ...**
>
> As the idea of exploiting jointly learnable priors by combining DDPMs and VAEs is quite general, we think it has the potential to be applied to other advanced models. However, in the current work we focus on signal restoration using standard conditional DDPMs. If we want to apply our idea to other variants, such as SDXL (Podell et al., 2024) that utilizes latent diffusion models (LDMs) (Rombach et al., 2022), more effort may be needed. To elaborate on it, note that our approach depends on the decent correlation between the target and the conditioning data. However, exploiting correlation in a more implicit domain, such as in the hidden space of an LDM, could still be possible while may also require more effort on studying and analyzing the characteristics of the signals in the hidden representation space. It might also call for proper design of specialized architectures or even loss functions for different encoders/decoders used in LDMs. Our current work has not addressed such aspects, but we think it will be very interesting to explore relevant research in future work.
>
> **Refs:**
> - Podell et al. SDXL: Improving latent diffusion models for high-resolution mage synthesis. In *ICLR*, 2024.
> - Rombach et al. High-resolution image synthesis with latent diffusion models. In *CVPR*, 2022.

---

### Decision · Program_Chairs · 2025-05-01

**Decision:**

Accept (poster)

**Comment:**

This paper seamlessly integrates DDPMs into the variational autoencoder framework and exploits the correlation between the degraded and clean signals to encode a better diffusion prior. The authors validate their method of speech enhancement and image restoration. This paper is reviewed by three experts in the field. The recommendations are: two weak accept and one accept. As a result, I think this paper can be accepted in the current form.